# Acetyltransferase Enok regulates transposon silencing and piRNA cluster transcription

**Shih-Ying Tsai**, **Fu Huang** *

Institute of Biological Chemistry, Academia Sinica, Taipei, Taiwan

* fuhuang@gate.sinica.edu.tw

## Abstract

The piRNA pathway is a highly conserved mechanism to repress transposon activation in the germline in *Drosophila* and mammals. This pathway starts from transcribing piRNA clusters to generate long piRNA precursors. The majority of piRNA clusters lack conventional promoters, and utilize heterochromatin- and HP1D/Rhino-dependent noncanonical mechanisms for transcription. However, information regarding the transcriptional regulation of piRNA clusters is limited. Here, we report that the *Drosophila* acetyltransferase Enok, which can activate transcription by acetylating H3K23, is critical for piRNA production from 54% of piRNA clusters including 42AB, the major piRNA source. Surprisingly, we found that Enok not only promotes *rhino* expression by acetylating H3K23, but also directly enhances transcription of piRNA clusters by facilitating Rhino recruitment. Taken together, our study provides novel insights into the regulation of noncanonical transcription at piRNA clusters and transposon silencing.

**Data Availability Statement:** All NGS files are available from the Gene Expression Omnibus database (accession number GSE105101).

## Author summary

Roughly half of our genome is composed of transposons. Activation of those transposons in the germline will result in severe DNA damages and infertility. The PIWI-interacting RNA (piRNA) pathway, which is highly conserved between mammals and flies, is a key mechanism to suppress transposon activation in the germline. Here, we identified the fly acetyltransferase Enok as a novel regulator functioning in the early steps of this pathway. We found that Enok can promote the expression of three genes involved in piRNA production by acetylating histone H3 lysine 23 (H3K23). We also demonstrated that Enok regulates the recruitment of Rhi, a factor critical for transcription initiation at piRNA-generating loci, to a subset of those loci, and therefore enhances their transcription. Our findings reveal an upstream regulator in the piRNA pathway and advance our understanding regarding the molecular mechanism of transposon silencing.

## Introduction

In a wide range of organisms, repressing the activation of transposon insertions is essential for maintenance of genome stability [1]. Small RNA-mediated heterochromatin formation plays

Processed data and values used in Figures are also listed in Tables S1–S7.

**Funding:** Acquisition of the small RNA-seq data was funded by the National Institutes of General Medical Sciences (https://www.nigms.nih.gov/) (grants RO1GM099945 and R35GM118068) to S. M. Abmayr and J.L. Workman. This study was funded by Academia Sinica (https://www.sinica.edu.tw/en) and the Ministry of Science and Technology (https://www.most.gov.tw/?l=de) (grant MOST105-2311-B-001-079-MY2) to F.H. The funders had no role in study design, data collection and analysis, decision to publish, or preparation of the manuscript.

**Competing interests:** The authors have declared that no competing interests exist.

important roles in silencing transposons in eukaryotic genomes [2]. Also, mammals and *Drosophila* utilize the PIWI-interacting RNA (piRNA) pathway to achieve transcriptional and post-transcriptional silencing of transposons in the germline [3,4]. In *Drosophila*, the piRNA pathway starts from transcription of the 142 piRNA clusters, usually ranging from 50 to a few hundred kilobases and containing multiple copies of truncated or full-length transposons, which produces long piRNA precursors [5–7]. The long RNA precursors would then be processed through slicer- and Zucchini (Zuc)-dependent mechanisms into mature 23–29 nt piRNAs that get loaded to the Piwi protein [8]. In addition, another two PIWI-clade Argonaute proteins, Ago3 and Aubergine (Aub), function in the ping-pong cycle to specifically amplify piRNAs against active transposons [9]. Guided by complementary piRNAs, Ago3 and Aub can mediate degradation of transposon transcripts, and the Piwi-piRNA complex can also direct heterochromatin formation at the loci of transposons and piRNA clusters by recruiting epigenetic factors [10–13], resulting in effective repression of transposons both transcriptionally and post-transcriptionally.

The *Drosophila* piRNA clusters can be divided into two classes: uni-strand, which produces piRNAs mainly from one genomic strand, and dual-strand, which produces piRNAs from both genomic strands. Transcription of the uni-strand clusters is proposed to be similar to the canonical transcription of protein-coding genes, as they contain clear promoter structures with enriched H3K4me2 and peaks of RNA polymerase II (Pol II) [14]. In contrast, dual-strand clusters lack clear Pol II promoter regions and are enriched for the heterochromatic H3K9me3 mark. Therefore, these clusters undergo noncanonical transcription that utilizes multiple internal initiation sites via heterochromatin- and Rhino (Rhi)-dependent mechanisms [6,15].

Rhi is the germline-specific heterochromatin protein 1D (HP1D), and it associates with Deadlock (Del) and Cutoff (Cuff) to form the RDC complex [14,16]. The RDC complex is recruited to dual-strand clusters by the interaction between H3K9me3 and the chromodomain of Rhi. At dual-strand clusters, the RDC complex licenses and promotes their transcription through four mechanisms. First, Del interacts with the germline-specific paralog of transcription factor IIA (TFIIA)-L, Moonshiner (Moon), and in turn recruits TFIIA and the TATA-box binding protein (TBP)-related factor TRF2 for transcription initiation [6]. Second, the RDC complex has been shown to suppress the splicing of piRNA cluster transcripts, which is proposed to facilitate piRNA production [16]. Third, Cuff recruits the transcription-export (TREX) complex to nascent transcripts to promote efficient transcription at piRNA clusters [17]. Fourth, Cuff interferes with recruitment of the cleavage and polyadenylation specificity factor (CPSF) complex, and therefore prevents premature termination during transcription of piRNA precursors [18]. While the positive roles of the RDC complex in noncanonical transcription of piRNA clusters were studied extensively, further transcriptional regulation upstream to the recruitment of this complex to piRNA clusters is still unclear.

The KAT6 acetyltransferases are highly conserved from budding yeast to mammals, and preferentially acetylate histone H3 among the four core histones [19]. The fly KAT6, Enok, has been shown to function as the major acetyltransferase for establishing the H3K23ac mark, which plays activating roles in transcription of genes [20,21]. H3K23ac has been suggested to destabilize the interaction between H3K27me3 and the chromodomain of Polycomb [22], and therefore may contribute to transcription activation. In the ovarian germline, Enok is important for the maintenance of germline stem cells [23], and is required for proper polarization of oocytes by promoting expression of the actin nucleator *spir* [20]. In this study, we report a novel role for Enok in the piRNA pathway. Mutating or knocking-down *enok* in the ovarian germline led to derepression of transposons and reduction in levels of piRNAs produced from a subset of piRNA clusters including the major piRNA source 42AB. We show that Enok binds

to and acetylates H3K23 in the 5' region of *rhi*, and is required for its normal expression levels in the ovary. We further show that Enok is also required for proper Rhi recruitment to a subset of piRNA clusters to promote their transcription. Therefore, Enok contributes to proper transposon silencing in the germline by promoting transcription of *rhi* and piRNA clusters.

## Results

### Transcript levels of 7 transposons are increased upon loss of functional Enok in the germline

The corresponding author has previously utilized the *FLP/FRT/ovoD* system to generate *enok* mutant germline clones to investigate the roles of Enok in the germline [20]. In this system, only egg chambers containing 16 homozygous mutant germline cells can develop beyond stage 4 of oogenesis [24]. Two *enok* mutant alleles, *enok¹* and *enok²*, were used in the germline clone analysis, and both alleles encode mutant Enok proteins with abolished acetyltransferase function [20]. The *enok¹* allele contains a W765-to-amber mutation, and produces a truncated Enok protein lacking more than 50% of the histone acetyltransferase (HAT) domain and the entire C-terminal region. The *enok²* allele contains a C746Y mutation inside the HAT domain that mutates a conserved cysteine residue, and therefore produces a full-length Enok protein with compromised HAT function. It was found that the *enok* mutant germline clone egg chambers showed defects in oocyte polarization. Therefore, RNA-seq analysis was performed using egg chambers of stages 5–14 from *enok* mutant or wild-type (WT) germline clone ovaries to identify genes involved in oocyte polarization that are transcriptionally regulated by Enok. Surprisingly, in addition to genes up- or down-regulated in *enok* mutants, we also found that several transposon families were activated in the *enok¹* and *enok²* germline clone ovaries as compared with the WT control (Fig 1). Given the high degree of correlation between each independent replicate in our next generation sequencing (NGS) data sets (S1, S2, and S3 Figs), results from one representative replicate are shown in all figures demonstrating a Genome Browser view. The transposon families activated upon loss of functional Enok in the germline include both the long terminal repeat (LTR) and the non-LTR transposons. For example, as shown in Fig 1A, the LTR transposon *Burdock* (left panel) and the non-LTR transposon *HeT-A* (right panel) were activated in both the *enok¹* and *enok²* mutants compared with WT ovaries. Furthermore, among the 126 transposon families, 7 families were significantly activated in *enok* mutant ovaries as compared with the WT control (Fig 1B and 1C, and S1 Table). Taking these results together, we concluded that Enok is important for proper silencing of 7 transposon families in the germline.

### Production of piRNAs from a subset of piRNA clusters is dependent on Enok

Since the piRNA pathway is the major mechanism to suppress transposon activation in the germline, we examined the piRNA levels in WT and the two *enok* mutant germline clone ovaries by small RNA-seq analysis (S2 Table). While the levels of piRNAs mapping to genic regions were similar in WT and *enok* mutants (Fig 2A; left panel), the levels of piRNAs that mapped to a subset of piRNA clusters were decreased in *enok* mutant ovaries as compared with the WT control (Fig 2A; middle and right panels). We further examined how many piRNA clusters were down- or up-regulated in *enok* mutants using DESeq2 (S2 Table). Because the levels of piRNAs uniquely mapping to cluster 70, 107, 110 and 128 were zero in WT samples, these four clusters were omitted, resulting in total 138 piRNA clusters in this analysis. Among these 138 piRNA clusters, 63 (46%) and 67 (49%) clusters showed

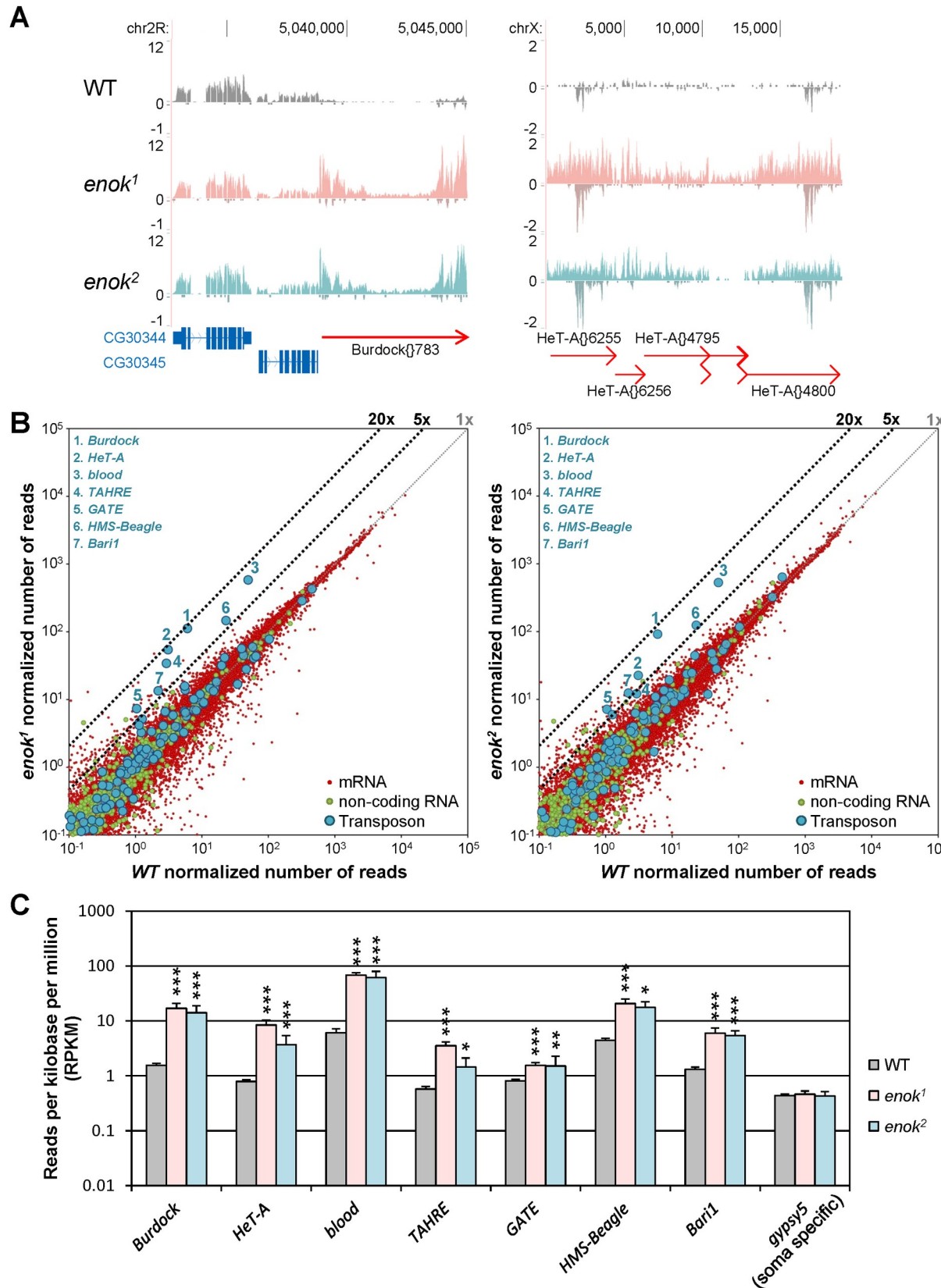

**Fig 1. Enok is important for transposon silencing in the germline. (A)** Genome Browser view of RNA-seq results near the LTR transposon *Burdock* and the non-LTR transposon *HeT-A* (red arrows) shows that both types of transposons were drastically activated in the *enok* mutant germline clone ovaries. However, expression levels of *CG30344* and *CG30345* (blue boxes) were unaltered. Reads per million (RPMs) are shown. **(B)** Seven transposon families were activated in the *enok* mutants as indicated by blue numbers. The levels of mRNAs, non-coding RNAs and transposons in *enok* mutant germline clone ovaries versus the WT control detected by RNA-seq are plotted in a log scale. **(C)** Seven transposon families were significantly up-regulated in *enok* mutants as compared with the WT control. Data represent the mean of 3 independent RNA-seq samples +/- SD. In (A-C), genotypes of females are as follows: *hs-Flp / +; FRT^G13^/ FRT^G13^, ovo^D1-18^* (WT); *hs-Flp / +; FRT^G13^, enok^1^/ FRT^G13^, ovo^D1-18^* (*enok^1^*); *hs-Flp/+; FRT^G13^, enok^2^/FRT^G13^, ovo^D1-18^* (*enok^2^*). \*adjusted P < 0.05, \*\* adjusted P < 0.01, \*\*\* adjusted P < 0.001 (DESeq2).

significantly decreased levels of piRNAs in both the *enok^1^* and *enok^2^* mutants compared with WT ovaries in the sense and antisense strands, respectively (S4A Fig). Also, 56 clusters (41%) showed reduced piRNA levels in both the sense and antisense strands in both *enok* mutants (S4A Fig; top right panel). In contrast, only 11 (8%) clusters had increased piRNA levels in both the sense and antisense strands in both *enok* mutants (S4A Fig; bottom right panel). Therefore, our data suggest that Enok mainly plays a positive role in the piRNA biosynthesis.

The major piRNA source 42AB and cluster 29 are among the most down-regulated clusters (Figs 2A and S4B). As shown in S5A Fig, levels of sense and antisense piRNAs mapping to 42AB were both reduced in *enok* mutants, and the sense piRNAs were affected to a larger extent than the antisense species. The ping-pong amplification of piRNAs mapping to 42AB was also compromised in *enok* mutant germline clone ovaries as compared with the WT control. We further examined the piRNAs mapping to transposons. While the *enok* mutants only showed mild reductions in antisense piRNAs mapping to all transposons (S5B Fig), antisense piRNAs mapping to the highly overexpressed *Burdock* transposon were more severely reduced (S5C Fig). This result suggests that our small RNA-seq data are in line with the results from RNA-seq analysis (Fig 1C) for a couple of transposons including *Burdock*.

After the piRNA clusters were first annotated [5], further studies have identified more regions in the fly genome that give rise to piRNAs [14]. Therefore, all the genomic regions giving rise to piRNAs were divided into 1 kb bins and defined as piRNA source loci (piRNA-SL). These piRNA-SL includes >91% of 1 kb bins defined by previously annotated piRNA clusters and extends the size of piRNA source regions more than 3-fold [14]. We further examined how mutations of *enok* affect levels of piRNAs mapping to the 3 types of piRNA-SL defined in the previous study [14]: Rhi-dependent (RD-SL), Rhi-independent (RI-SL) and soma-specific (SO-SL). A population of heterochromatic 1 kb bins not giving rise to piRNAs (het non-SL) was also included as the control group [14]. As shown in Fig 2B, levels of piRNAs mapping to RD-SL were significantly decreased in the two *enok* mutants as compared with the WT control. In contrast, piRNAs mapping to RI-SL, SO-SL or het non-SL were not down-regulated in *enok* mutants (Fig 2B). Therefore, Enok may play an important role in the production of piRNAs that are enriched in germline cells and dependent on Rhi.

Next, we determined the respective Enok dependency (piRNA fold change upon *enok* mutation) of each RD-SL (S6A Fig) [14]. Based on this dependency, the majority (87%) of RD-SL were split into two populations. The log2 values of piRNA fold changes from the first population (n = 1074) were less than -1.5 in both the *enok^1^* and *enok^2^* mutants as compared with the WT control (S6A and S6B Fig), and therefore piRNAs from this population were dependent on Enok (Enok-dependent source loci; ED-SL). The second population (n = 4501) produces piRNAs that were not or only weakly dependent on Enok (Enok-independent source loci; EI-SL) (S6A Fig). Notably, only 404 RD-SL showed log2 values of piRNA fold changes more than 1.5 in both *enok* mutants compared with the WT control (S6C Fig), again suggesting that Enok mainly plays a positive role in piRNA production. As shown in Fig 2C, piRNAs

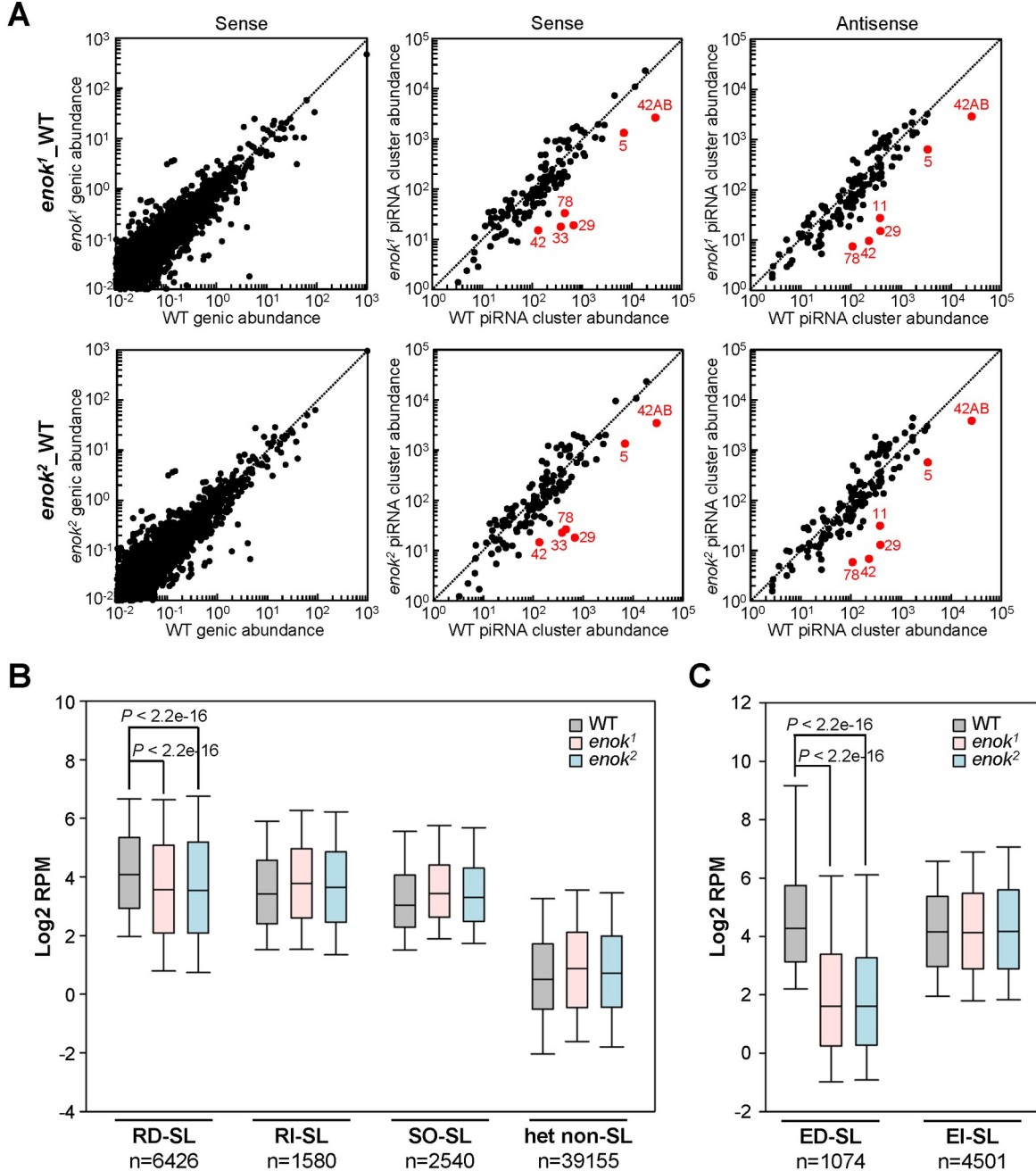

**Fig 2. Enok is important for piRNA production from a subset of piRNA clusters. (A)** Loss of functional Enok in the germline reduced levels of piRNAs from a subset of piRNA clusters but not from genic regions. The levels of piRNAs that uniquely mapped to genic regions (left panel) and piRNA clusters (middle and right panels) in *enok* mutant germline clone ovaries versus the WT control are plotted in a log scale. Selected piRNA clusters that were severely down-regulated in *enok* mutants are indicated by red dots. **(B-C)** A box-plot displaying the distribution of piRNA levels for all 1 kb bins belonging to the indicated groups. Center line, median; box limits, upper and lower quartiles; whiskers, the 5th and 95th percentile (outliers not shown). In (A-C), genotypes of females are as described in Fig 1. In (B-C), P-values were calculated using Wilcoxon Rank sum test.

from ED-SL were severely reduced in both *enok* mutants, while EI-SL piRNAs were unaffected. Thus, we concluded that Enok may be important for piRNA production from ~20% of the Rhi-dependent loci (S3 Table).

## Enok regulates the expression levels of three genes involved in piRNA biosynthesis

We sought to investigate the mechanisms by which Enok contributes to the Rhi-dependent piRNA production. First, we examined the expression levels of genes known to be involved in the piRNA pathway. A previous transcriptome-wide RNAi screen has identified top 100 genes that are important for the germline piRNA pathway [25]. Among them, 9 genes showed a RPKM value less than 5 in our RNA-seq data obtained from WT ovaries, and therefore were excluded from the gene list. In the remaining 91 genes, expression levels of 5 genes, *rhi*, *Brother of Yb* (*BoYb*), *maelstrom* (*mael*), *shutdown* (*shu*) and *CG12721/moon*, were significantly down-regulated in the *enok* mutant germline clone ovaries as compared with the WT control (Figs 3A and S7A). In contrast, levels of other known key regulators of the piRNA pathway, including *piwi*, were largely unaffected upon loss of Enok in the germline (S7A and S7B Fig, and S4 Table). Immunostaining in the *enok* mutant germline clone ovaries also revealed reductions in the protein levels of Rhi and Mael (S7C and S7D Fig) similar to those in their mRNA levels (S7A Fig). However, the *enok* mutant ovaries still expressed more than 47% of the wild-type levels of *BoYb*, *mael*, *shu* and *moon* mRNA, which is similar to or better than a haploid condition. On the other hand, the expression levels of *rhi* showed a ~75% reduction in *enok* mutants as compared with WT (Fig 3A and S4 Table). Since heterozygotes of most piRNA pathway components show normal transposon silencing in the germline [26], our results suggest that Enok may contribute to proper piRNA biosynthesis partly by promoting the expression of *rhi*. Nevertheless, we cannot exclude the possibility that the combination of those mild reductions in *BoYb*, *mael*, *shu* and *moon* levels may also have effects on piRNA production.

To examine whether *enok* and *rhi* mutants show similar effects on piRNA production, we first analyzed the abundance of antisense piRNAs encoded by each transposon family (S8A Fig). Among the 7 transposon families activated in *enok* mutants (Fig 1C), levels of antisense piRNAs encoded by *Burduck*, *HeT-A* and *Bari1* were reduced in *enok* mutants as compared with the WT control, while the other 4 transposon families only showed subtle decreases in antisense piRNAs (S8A Fig). The decreases in levels of primary piRNAs in *enok* mutants may be masked by ping-pong mechanism with participation of maternal transposon-specific piRNAs, and the increased RNA levels of *blood*, *TAHRE*, *GATE* and *HMS-Beagle* in *enok* mutants (Fig 1C) may be determined by another mechanism. We further analyzed the fold change in transposon family expression in *enok* mutants (S8B Fig; y axis, enok1/WT or enok2/WT) and the fold change in antisense piRNAs encoded by the same transposon family (S8B Fig; x axis, enok1/WT or enok2/WT) as performed in the previous study for the *rhi* mutant [15]. The results for *enok* mutants (S8B Fig) show some similarity to those for the *rhi* mutant [15]: some of the highly overexpressed transposons (*Burdock* and *HeT-A*) also showed large reductions in antisense piRNAs, while expression of *blood* increased 10–12 fold but the total antisense piRNA pool was reduced by only 12%-25%. This similarity between *enok* and *rhi* mutants further supports the hypothesis that Enok may promote piRNA biosynthesis by regulating *rhi* expression.

To further examine whether Enok binds to and directly regulates transcription at these 5 genes involved in piRNA biosynthesis, Enok ChIP-seq analysis was performed in wild-type ovaries using two independently raised α-Enok antibodies (S9A Fig). ChIP-seq results showed that Enok was localized to the 5' regions of *rhi*, *mael* and *shu* (Figs 3B and S9B). This result is also in line with the previous report that Enok was enriched at the 5' end of *mael* and depleting Enok in S2 cells reduced Pol II occupancy at *mael* 5' [20]. Furthermore, ChIP-qPCR analysis showed that Enok levels at the 5' end of *rhi*, *mael* and *shu* were reduced upon germline-specific knockdown of *enok* (Fig 3C). Therefore, our results suggest that Enok may contribute to

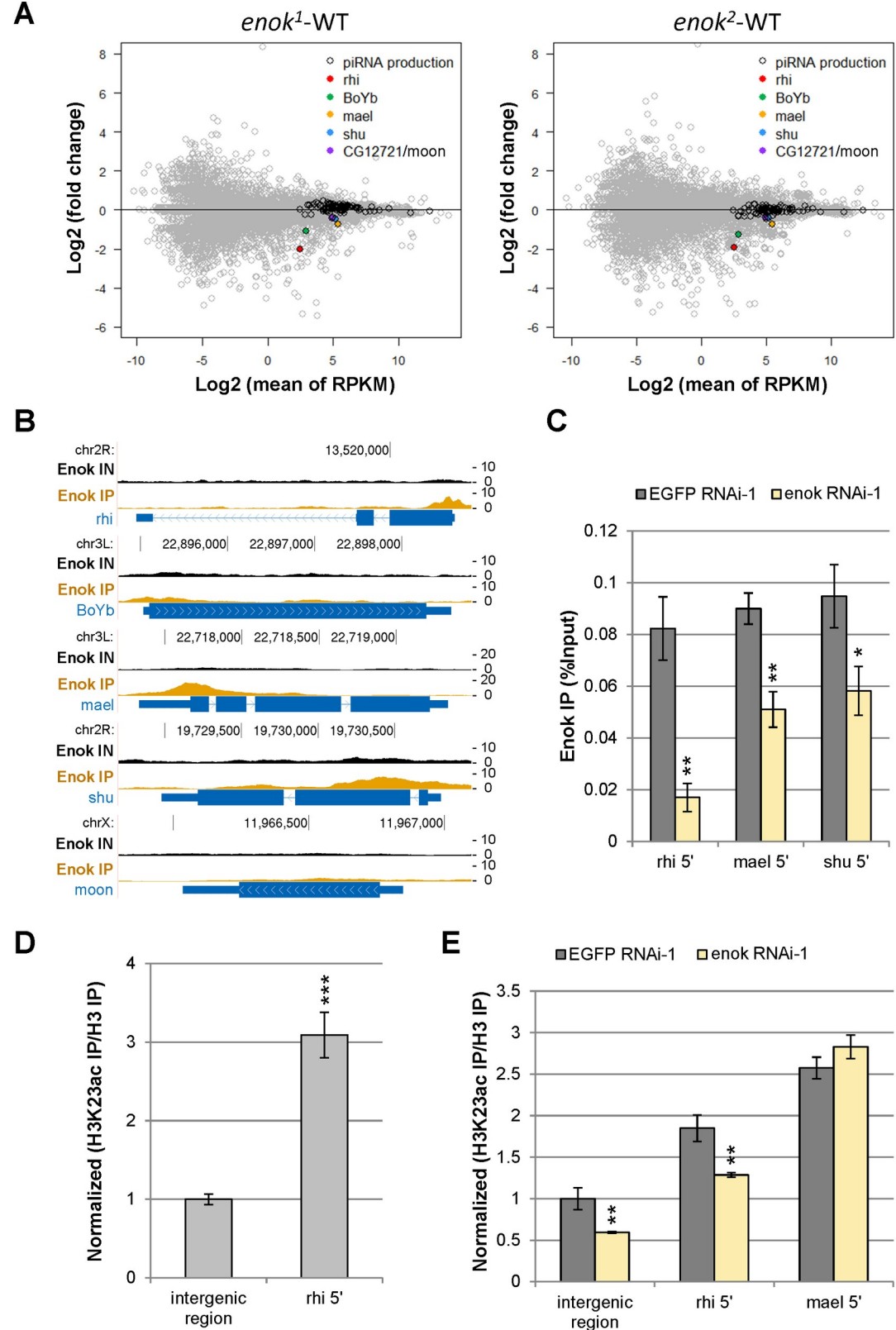

**Fig 3. Enok promotes expression of genes involved in piRNA biosynthesis by acetylating H3K23. (A)** *rhi*, *BoYb*, *mael*, *shu* and *CG12721/moon* were down-regulated in the *enok* mutant germline clone ovaries. MA plots showing fold change in

expression plotted against the expression levels for the *enok¹* (left panel) or the *enok²* (right panel) mutant ovaries versus the WT control. RPKM, reads per kilobase per million. Ninety-one genes (RPKM > 5 in WT) involved in the piRNA pathway are marked with black circles. Genotypes of females are as described in Fig 1. **(B)** Genome Browser view of Enok ChIP-seq data at *rhi*, *BoYb*, *mael*, *shu* and *CG12721/moon*. ChIP-seq experiments were performed in ovaries from Oregon-R (*OreR*) in two independent replicates, and results from one representative replicate are shown. Input (IN) and Immunoprecipitation (IP) tracks are shown. Enrichment of Enok was observed in the 5' region of *rhi*, *mael* and *shu*. The FlyBase Genes track is shown to represent the genes, showing the exon (wider bars) and intron (blue line) structure of each annotated coding transcript, with direction of transcription indicated. **(C)** ChIP-qPCR analysis of Enok was performed in ovaries with the control (EGFP RNAi-1) or *enok* knockdown. Genotypes are as follows: *P{w[+mC] = otu-GAL4::VP16.R}1 / +; P{w[+mC] = GAL4-nos.NGT}40 / +; P{w[+mC] = GAL4::VP16-nos.UTR}CG6325^{MVD1} / P{VALIUM20-EGFP.shRNA.1}attP2* (EGFP RANi-1); *P{w[+mC] = otu-GAL4::VP16.R}1 / +; P{w[+mC] = GAL4-nos.NGT}40/+; P{w[+mC] = GAL4::VP16-nos.UTR}CG6325^{MVD1} / P{TRiP.HMS02634}attP2* (enok RNAi-1). **(D)** The H3K23ac levels at the 5' region of *rhi* (rhi 5') in the ovary of *OreR* were analyzed by ChIP-qPCR. The H3K23ac IP signals were first normalized to the histone H3 IP signals (H3K23ac/H3), and then the H3K23ac/H3 values were normalized to the mean of H3K23ac/H3 values obtained for the intergenic region, which was set as 1. **(E)** The H3K23ac levels in the 5' regions of *rhi* (rhi 5') and *mael* (mael 5') in ovaries were analyzed by ChIP-qPCR. The H3K23ac IP signals were first normalized to the histone H3 IP signals (H3K23ac/H3), and then the H3K23ac/H3 values were normalized to the mean of H3K23ac/H3 values obtained for the intergenic region in EGFP RNAi-1 ovaries, which was set as 1. Genotypes of the females are as follows: *tj-GAL4/+; P{w[+mC] = GAL4::VP16-nos.UTR}CG6325^{MVD1} / P{VALIUM20-EGFP.shRNA.1}attP2* (EGFP RNAi-1); *tj-GAL4/+; P{w[+mC] = GAL4::VP16-nos.UTR}CG6325^{MVD1}/P{TRiP.HMS02634}attP2* (enok RNAi-1). In (C-E), the amplicon for intergenic region is located at chr2L:18,415,855–18,415,978. The amplicon for *rhi* 5', *mael* 5' or *shu 5'* is located at the position -62-96, 14–179 or 1–153 relative to the transcription start site of *rhi*, *mael* or *shu*, respectively. Data represent the mean of three biological replicates +/- SD. *P < 0.05, **P < 0.01, ***P < 0.001 (Student's t-test).

proper piRNA production partly by binding to *rhi*, *mael* and *shu* and facilitating their transcription.

It was previously demonstrated that Enok is the major enzyme for establishing the H3K23ac mark in *Drosophila*, and the Enok-mediated H3K23ac mark plays positive roles in transcriptional regulation [20]. Therefore, we asked whether H3K23ac is enriched at *rhi* and *mael*. To answer this question, ChIP-qPCR analysis of H3K23ac was carried out in the ovary. As shown in Fig 3D and 3E, the H3K23ac mark was enriched 2–3 fold in the 5' regions of *rhi* and *mael* relative to the intergenic region. Moreover, knocking down *enok* in ovaries reduced the H3K23ac level in the 5' region of *rhi* (Fig 3E), supporting that Enok acetylates H3K23 at *rhi* 5'. On the other hand, since the Enok level at mael 5' only reduced 43% upon *enok* knockdown (Fig 3C), the H3K23ac level at *mael* 5' did not reduce in *enok* knockdown ovaries (Fig 3E). However, it was previously shown that depleting Enok in S2 cells reduced H3K23ac levels in the 5' and 3' regions of *mael* [20]. Taken together, we concluded that Enok may promote the expression of *rhi* and *mael* by acetylating H3K23.

Rhi is the germline-specific HP1D. It forms the RDC complex with Del and Cuff, and is recruited to the H3K9me3-enriched dual-strand piRNA clusters to license noncanonical transcription and to suppress piRNA precursor splicing [6,16]. Since *rhi* is severely down-regulated in *enok* mutant ovaries, we asked whether levels of RNAs derived from the Rhi-dependent 42AB and the Rhi-independent cluster 2 are regulated by Enok. To answer this question and to verify our RNA-seq results obtained from *enok* mutant germline clone ovaries, RNAi against *enok* in the ovarian germline was carried out using two different *UAS-shRNA-enok* transgenes driven by MTD-Gal4 (*P{w[+mC] = otu-GAL4::VP16.R}1; P{w[+mC] = GAL4-nos.NGT}40; P{w[+mC] = GAL4::VP16-nos.UTR}CG6325^{MVD1}*). Then, the RNA levels in the *enok* knockdown and control ovaries were examined by RT-qPCR. Consistent with the RNA-seq results in *enok* mutants, knocking down *enok* in the germline resulted in activation of the *Burdock* transposon as compared with the *EGFP RNAi-1* control (S10A Fig). Also, the *rhi* mRNA levels were significantly reduced in both the *enok RNAi-1* and *enok RNAi-2* ovaries when RNAi against *enok* was performed at 29°C for 3 days (S10A Fig), suggesting that Enok indeed regulates the expression of *rhi*. The levels of RNAs derived from 42AB and cluster 2 were further examined by strand-specific RT-qPCR. Similar to the *rhi* mutant [15], depletion of Enok

in the germline led to decreased levels of RNAs derived from 42AB but increased levels of RNAs from cluster 2 (S10B Fig). This result suggests that Enok is important for transcription of the Rhi-dependent 42AB cluster.

We further analyzed the RNA-seq reads uniquely mapping to piRNA clusters to examine whether levels of RNAs transcribed from each cluster were decreased in the *enok* mutant germline clone ovaries. Six piRNA clusters (42AB, cluster 5, 29, 78, 131 and 137) were significantly down-regulated (adjusted *P*-value < 0.05) in the two *enok* mutant ovaries (S5 Table). Also, as shown in S11 Fig, the four severely down-regulated clusters in RNA-seq analysis (42AB, cluster 5, 29 and 78; red dots) match the severely down-regulated clusters identified by small RNA-seq (Fig 2A; red dots). On the other hand, the RNA-seq reads uniquely mapping to three severely down-regulated clusters labeled in Fig 2A (clusters 11, 33 and 42) were not significantly affected. Cluster 11 contains very low levels of unique RNA-seq reads, and this may be why the adjusted p-value is not significant (>0.05; S5 Table). Clusters 33 and 42 both contain an expressed gene within them, and therefore decreases in the relatively low levels of piRNA precursor transcripts may be masked by the mRNA levels from those two genes. Taken together, these results suggest that the defective transcription of a subset of piRNA clusters in *enok* mutants indeed resulted in reduced piRNA production from those clusters.

## Enok has roles in promoting transcription of piRNA clusters that are independent of its role in regulating rhi expression

We noticed that the *rhi* mRNA level in the *enok RNAi-1* ovaries was reduced only ~30% (S10A Fig), suggesting that the effects on RNA levels transcribed from 42AB in the *enok RNAi-1* ovaries may be independent of the role of Enok in promoting *rhi* expression. To test this possibility, RNAi against *enok* was carried out at 25˚C for 1 day and then at 29˚C for 2 days to knockdown *enok* more mildly, as the activity of Gal4 is higher at 29˚C than 25˚C. Under this condition, we were able to obtain efficient *enok* knockdown without affecting the mRNA levels of *rhi*, *mael* and *BoYb* in the *enok RNAi-1* ovaries (Fig 4A). Supporting our hypothesis, strand-specific RT-qPCR analysis showed that levels of RNAs derived from 42AB, except for cl1-32-minus, were significantly decreased in the *enok RNAi-1* ovaries as compared with the *Luciferase* control (Fig 4B; left panel). On the other hand, levels of RNAs derived from cluster 2 were not decreased upon depletion of Enok (Fig 4B; right panel). Since knocking down *enok* in the germline without affecting *rhi* levels decreased the levels of RNAs from 42AB, our results suggest that, in addition to its role in regulating *rhi* levels, Enok may play another role in facilitating transcription of piRNA clusters including 42AB.

We further examined whether knocking down *enok* in the germline can indeed reduce the levels of RNAs derived from 42AB without affecting the bulk protein levels of Rhi. As shown in S10A Fig, even when RNAi against *enok* was performed at 29˚C for 3 days, the mRNA levels of *rhi* and *mael* in the *enok RNAi-1* ovaries were only decreased 28% and 13%, respectively. Consistent with these small changes in the *rhi* and *mael* levels, immunostaining showed only mild reductions in staining signals of Mael and Rhi in the *enok RNAi-1*ovaries as compared with the *EGFP RNAi-1* control (S12A and S12B Fig). Since the staining signals of endogenous Rhi were relatively weak and the α-Rhi antibody does not work for western blotting, we further utilized the *GFP-rhi* transgene expressing GFP-tagged Rhi in the context of its endogenous control region as reported in the previous study [14]. Both western blotting and fluorescence microscopy showed that the global levels of GFP-Rhi were largely unaffected upon germline-specific knockdown of *enok* as compared with the control (*Luciferase*) (S12C–S12E Fig). Also,

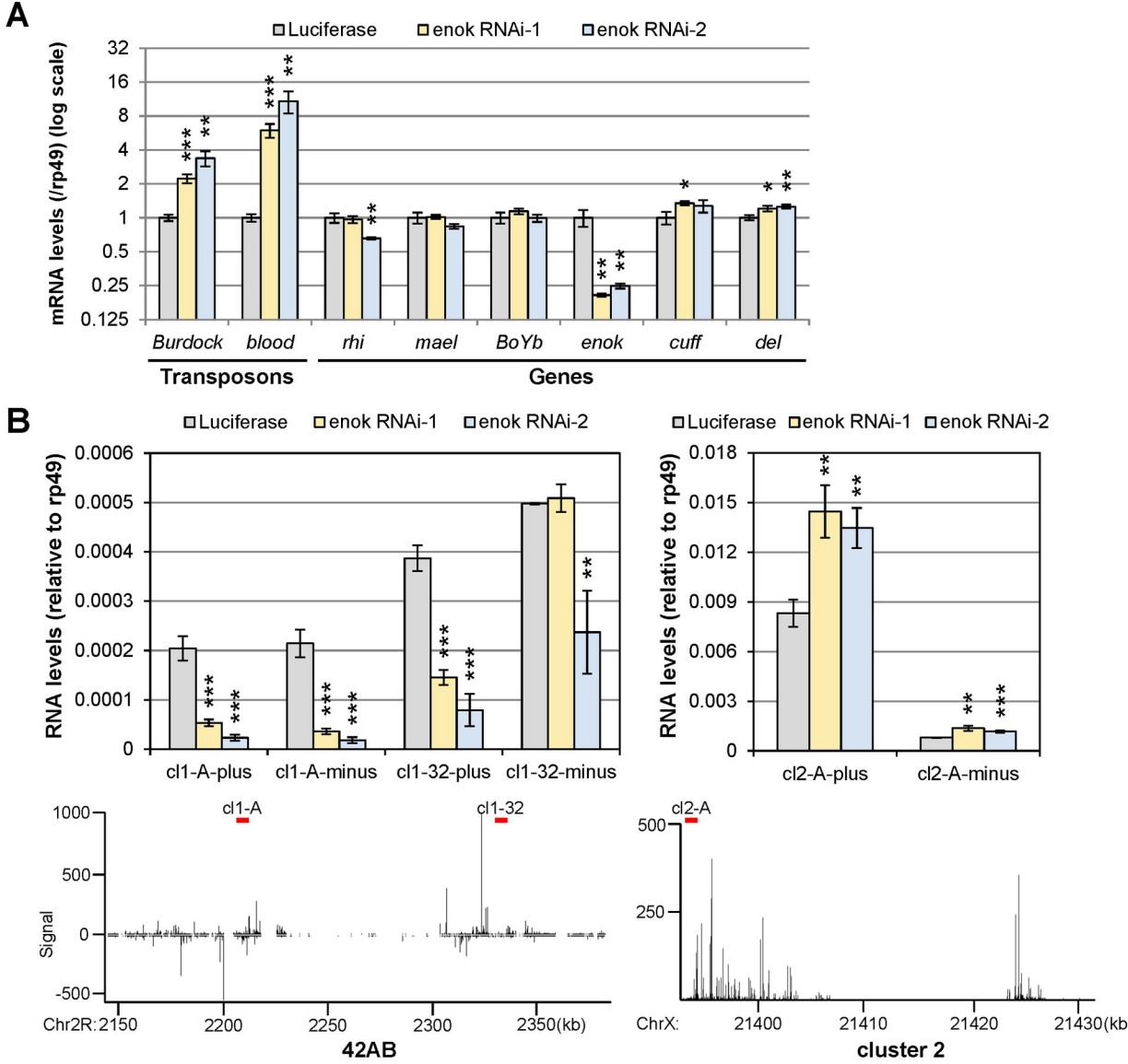

**Fig 4. Knocking down *enok* without affecting *rhi* levels decreased transcript levels from 42AB. (A)** Transposons were activated upon *enok* knockdown without affecting *rhi* levels. RT-qPCR analysis of ovaries was used to examine expression levels of the indicated transposons and genes. The mRNA levels were normalized to the levels of *rp49*. **(B)** Levels of RNAs from 42AB were reduced upon *enok* knockdown without affecting *rhi* levels. The same total RNA samples used in (A) were subjected to strand-specific RT-qPCR analysis for RNAs derived from dual-strand cluster 42AB (left panel) and uni-strand cluster 2 (right panel). The bottom panel shows the signals of piRNA reads mapping to the indicated clusters in WT ovaries as described in Fig 2. Red bars in the bottom panel indicate the location of amplicons used in the qPCR reaction. In (A-B), data represent the mean of three biological replicates +/- SD. *P < 0.05, **P < 0.01, ***P < 0.001 (Student's t-test). Genotypes of the females are as follows: *P{w[+mC] = otu-GAL4::VP16.R}1 / +; P{w[+mC] = GAL4-nos.NGT}40 / +; P{w[+mC] = GAL4::VP16-nos.UTR} CG6325^{MVD1} / P{UAS-LUC.VALIUM10}attP2* (Luciferase); *P{w[+mC] = otu-GAL4::VP16.R}1 / +; P{w[+mC] = GAL4-nos.NGT}40 / +; P{w[+mC] = GAL4::VP16-nos.UTR}CG6325^{MVD1} / P{TRiP.HMS02634}attP2* (enok RNAi-1); *P{w[+mC] = otu-GAL4::VP16.R}1 / +; P{w[+mC] = GAL4-nos.NGT}40/+; P{w[+mC] = GAL4::VP16-nos.UTR}CG6325^{MVD1}/P{TRiP.HMS02048}attP2* (enok RNAi-2).

bulk levels of the GFP-tagged Del, which forms a complex with Rhi [14], were similar in the *Luciferase* and *enok RNAi-1* ovaries (S12C Fig; lanes 4–6). Taken together, we concluded that knocking down *enok* in the germline can decrease levels of RNAs from 42AB (S10B Fig; *enok RNAi-1*) without affecting the global Rhi levels (S12B–S12E Fig).

## Enok is important for Pol II occupancies at ED-SL

To further test the hypothesis that Enok may regulate transcription at piRNA clusters, ChIP-seq analysis of Pol II was performed in ovaries. The Pol II ChIP-seq was carried out in ovaries with the *UAS-EGFP RNAi-1* (GFP_KD) or *UAS-enok RNAi-1* (*enok*_KD) transgene driven by MTD-Gal4 to examine whether Enok is important for Pol II occupancies at piRNA clusters. As shown in Fig 5A, while sites of Pol II occupancy at cluster 2 were not affected upon depletion of Enok in the germline (right panel), those at 42AB were decreased in the *enok*_KD ovaries compared with the GFP_KD control in two independent replicates (left panel; light yellow shades). This result suggests that Enok has a role in promoting transcription of a subset of dual-strand clusters including 42AB.

We further examined the changes in Pol II occupancies at ED-SL upon *enok* knockdown. Consistent with our small RNA-seq results (Fig 2C), Pol II occupancies at ED-SL were significantly reduced upon *enok* knockdown, while those at EI-SL were unaffected (Fig 5B). In addition, RNA-seq reads uniquely mapping to ED-SL were also significantly decreased in the two *enok* mutants (Fig 5C). Since knocking down *enok* without affecting Rhi levels (S12C–S12E Fig) decreased Pol II occupancies at ED-SL (Fig 5B), we concluded that Enok may have a role, which is independent of its role in promoting *rhi* expression, in enhancing transcription of these Enok-dependent loci.

## Enok is critical for the recruitment of Rhi to ED-SL

Next, we sought to investigate how Enok regulates transcription of piRNA clusters. As Enok ChIP-seq analysis did not detect Enok peaks at 42AB (Fig 5A) or enrichment of Enok at ED-SL relative to EI-SL (S13A Fig), this regulation is unlikely to be mediated by the H3K23ac mark. H3K23 and H3 ChIP-seq analyses in GFP_KD and *enok*_KD ovaries further confirmed that the H3K23ac levels mapping to piRNA clusters or transposons were not decreased upon *enok* knockdown (S13B Fig and S6 Table). Therefore, we explored other possibilities. Rhi anchors the RDC complex at piRNA clusters by recognizing the H3K9me3 mark, and has been reported to play multiple key roles in seeding and facilitating productive transcription at all dual-strand piRNA clusters [6,14–18]. Although the global protein levels of Rhi were largely unaffected upon *enok* knockdown (S12C–S12E Fig), it is possible that Enok may regulate Rhi recruitment to Enok-dependent piRNA clusters. Indeed, knocking down *enok* in the germline severely reduced both the transcript levels and Rhi occupancy at 42AB (Figs 6A, S14A and S14B). Since Rhi recognizes H3K9me3, we further examined the effects of *enok* knockdown on the H3K9me3 levels at 42AB by ChIP analysis. Upon depletion of Enok in the germline, levels of H3K9me3 at the cl1-A and cl1-32 sites in 42AB were reduced only 10%-20% as compared with the control *EGFP RNAi-1* ovaries (S14C Fig). It has been shown previously that the H3K9me3 levels were decreased ~65% at the cl1-A site when *piwi* was knocked down during embryogenesis, and the Rhi levels at the same site were also reduced ~65% [12]. Therefore, the ~85% reduction in Rhi occupancy at 42AB upon *enok* knockdown (Fig 6A; cl1-A) is unlikely to be mediated by the ~15% reduction in H3K9me3 levels at 42AB (S14C Fig; cl1-A). Also, H3K9me3 levels in the 5' regions of *Burdock* and *HeT-A* remained similar in the *EGFP RNAi-1* and *enok RNAi-1* ovaries (S14C Fig). These results suggest that depletion of Enok may not have great impacts on H3K9me3 levels at piRNA clusters or transposons.

To further assess the effects of *enok* knockdown on Rhi recruitment to piRNA clusters/ source loci, Rhi ChIP-seq analysis was performed in the GFP_KD and *enok*_KD ovaries. Consistent with the Rhi ChIP-qPCR result (Fig 6A), knocking down *enok* in the germline resulted in decreased Rhi occupancies across 42AB (Fig 6B). Also, Rhi occupancies at ED-SL were significantly reduced upon *enok* knockdown, while those at EI-SL were unaffected (Fig 6C). As

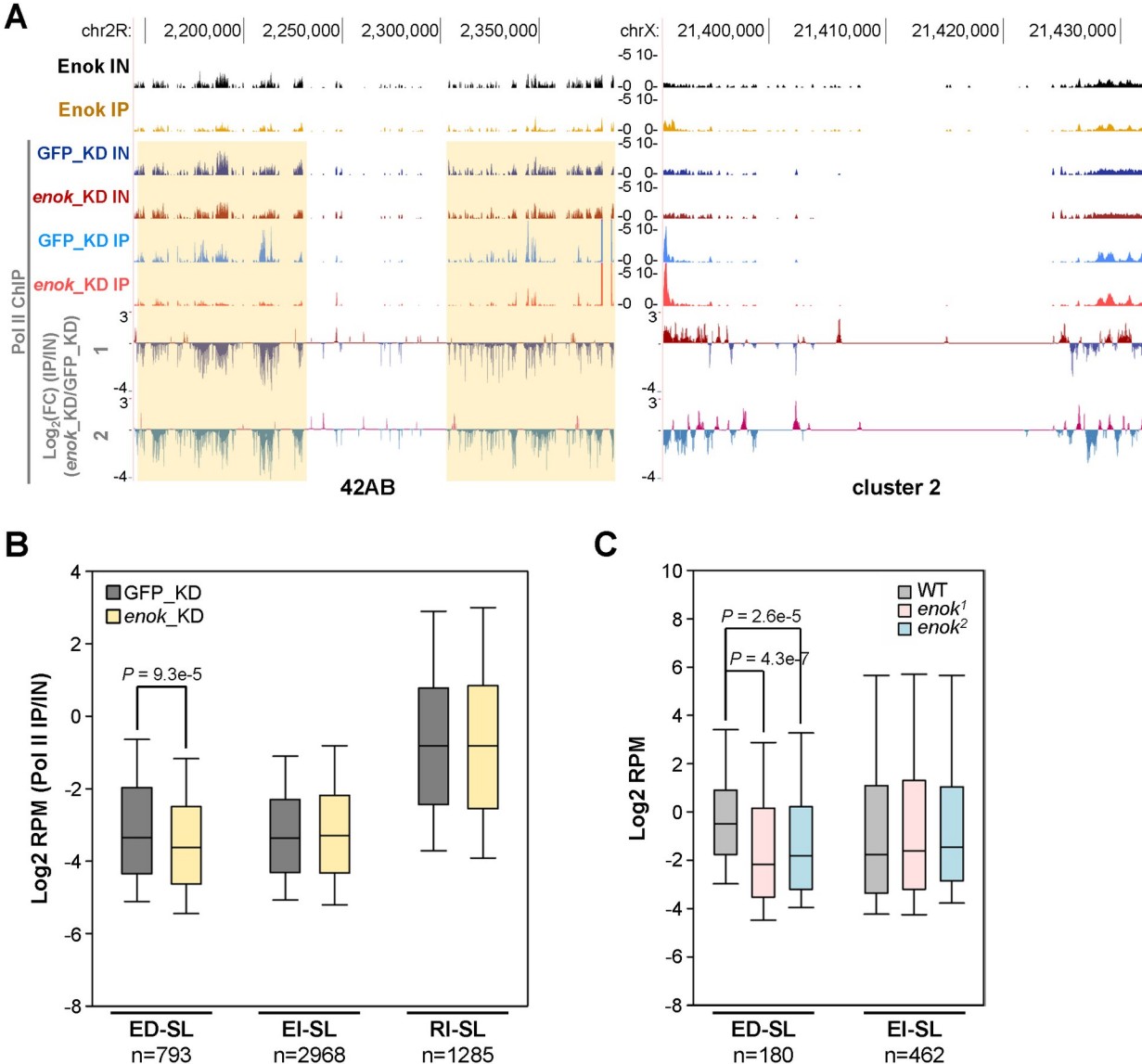

**Fig 5. Enok is important for transcription at Enok-dependent piRNA source loci. (A)** Genome Browser view of Enok and Pol II ChIP-seq data at 42AB and cluster 2 shows that the Pol II levels at 42AB were decreased upon *enok* knockdown. ChIP-seq experiments were performed in two independent replicates, and results from one representative replicate are shown. Input (IN) and immunoprecipitation (IP) tracks are shown in the top 6 panels. The bottom two panels show the $\log_2$ value of fold change (FC) in Pol II occupancy in *enok* knockdown ovaries (*enok*_KD) over the control knockdown (GFP_KD) from two independent replicates, with normalization to input. Light yellow shades indicate regions with reproducible decreased Pol II occupancy in *enok*_KD as compared with GFP_KD. Enok ChIP-seq experiments were performed in ovaries from *OreR*. **(B)** A box-plot displaying the distribution of Pol II occupancies for all 1 kb bins belonging to the indicated groups. Center line, median; box limits, upper and lower quartiles; whiskers, the 5th and 95th percentile (outliers not shown). **(C)** A box-plot displaying the distribution of RNA levels for all 1 kb bins belonging to the indicated groups. Center line, median; box limits, upper and lower quartiles; whiskers, the 5th and 95th percentile (outliers not shown). Genotypes of females are as described in Fig 1. In (A-B), genotypes of the GFP_KD and *enok*_KD females are as follows: *P{w[+mC] = otu-GAL4::VP16.R}1 / +; P{w[+mC] = GAL4-nos.NGT}40 / +; P{w[+mC] = GAL4::VP16-nos.UTR}CG6325^MVD1 / P {VALIUM20-EGFP.shRNA.1}attP2* (GFP_KD); *P{w[+mC] = otu-GAL4::VP16.R}1 / +; P{w[+mC] = GAL4-nos.NGT}40/+; P{w[+mC] = GAL4:: VP16-nos.UTR}CG6325^MVD1/P{TRiP.HMS02634}attP2* (*enok*_KD). In (B-C), *P*-values were calculated using Wilcoxon Rank sum test.

the global protein levels of Rhi were largely unaffected in *enok* knockdown ovaries (S12C-S12E Fig), this result suggests that Enok may promote transcription of these Enok-dependent piRNA source loci by facilitating Rhi recruitment.

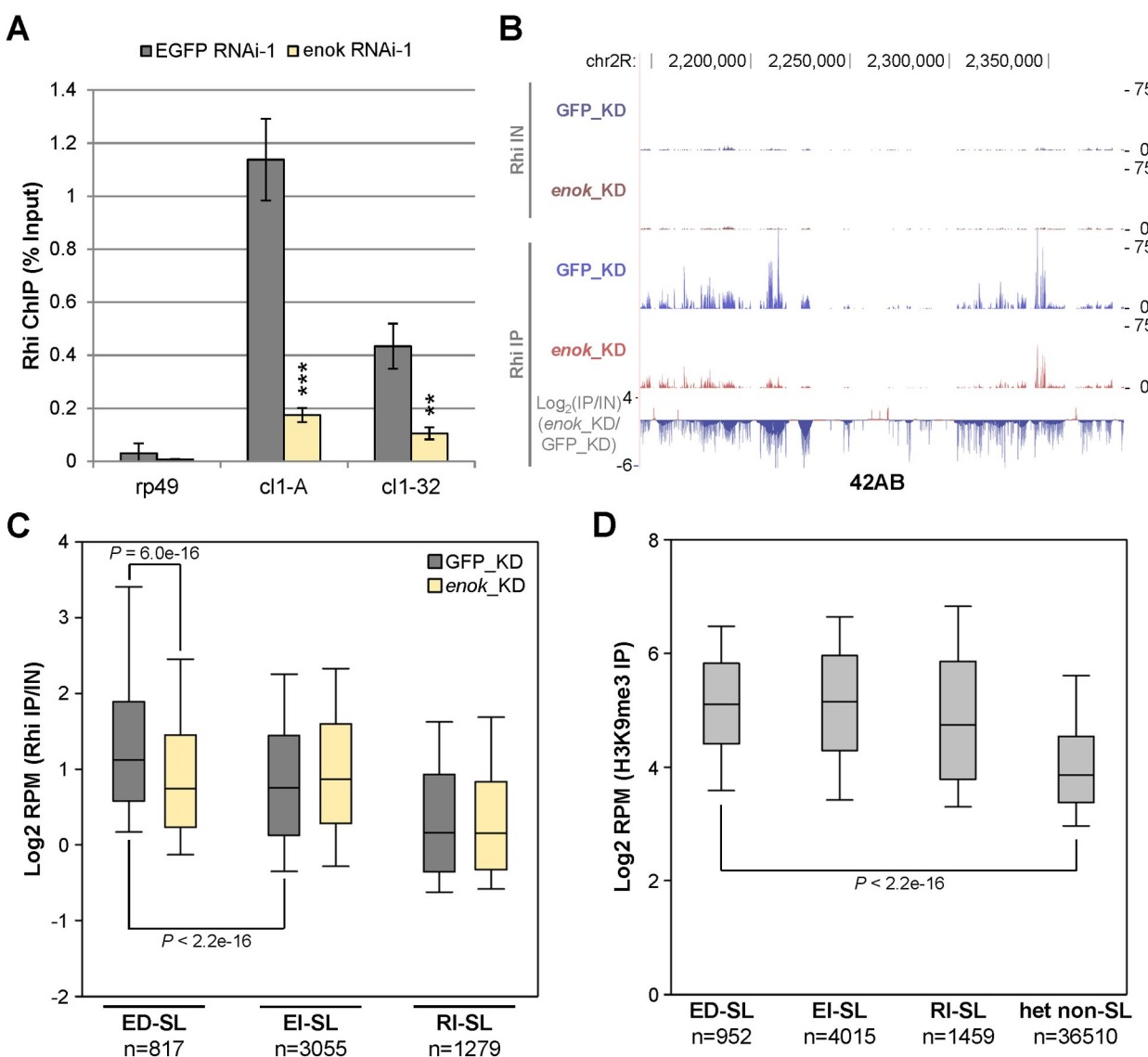

**Fig 6. Enok is important for Rhi occupancies at Enok-dependent piRNA source loci.** **(A)** Enok is important for Rhi recruitment to 42AB. The Rhi levels at 42AB in ovaries with germline-specific control knockdown (EGFP RNAi-1) or *enok* knockdown were analyzed by ChIP-qPCR. The location of amplicons used for 42AB is the same as indicated in Fig 4B. Genotypes of the females are as described in Fig 3C. **P < 0.01, ***P < 0.001 (Student's t-test). **(B)** Genome Browser view of Rhi ChIP-seq data shows that the Rhi levels at 42AB were decreased upon *enok* knockdown. Experimental details are as described in Fig 5A. The bottom panel shows the log$_2$ value of fold change in Rhi occupancy in *enok* knockdown ovaries (*enok*_KD) over the control knockdown (GFP_KD), with normalization to input. **(C)** A box-plot displaying the distribution of Rhi occupancies for all 1 kb bins belonging to the indicated groups. Center line, median; box limits, upper and lower quartiles; whiskers, the 5th and 95th percentile (outliers not shown). **(D)** A box-plot displaying the distribution of H3K9me3 levels for all 1 kb bins belonging to the indicated groups. Center line, median; box limits, upper and lower quartiles; whiskers, the 5th and 95th percentile (outliers not shown). In (B-C), genotypes are as described in Fig 5. In (C-D), *P*-values were calculated using Wilcoxon Rank sum test.

Notably, Rhi occupancies at ED-SL were higher than those at EI-SL in the control knockdown ovaries (Fig 6C; GFP_KD). However, the H3K9me3 levels (GEO accession GSE55824) were similar between ED-SL and EI-SL (Fig 6D). Therefore, Enok may have a role in enhancing Rhi recruitment/binding to the Enok-dependent loci as compared with the Enok-independent loci.

Since knocking down *enok* in the germline led to reduced Rhi occupancy at 42AB, we sought to examine whether overexpression of *rhi* can rescue this defect in the *enok* knockdown ovaries. To this end, the *UAS-RNAi* or *UAS-Luciferase* construct and an *UASp-rhi:GFP* transgene [15] were overexpressed in the ovarian germline by the nos-Gal4-VP16 driver (S15A and S15B Fig). Interestingly, while the *UASp-rhi:GFP* transgene was overexpressed up to ~64-fold of the endogenous *rhi* levels (Figs 7A and S15C), the defective transcription at 42AB in the *enok RNAi-1* ovaries could not be rescued by overexpressing *GFP-rhi* (Figs 7B and S15D). Moreover, ChIP analysis of Rhi showed that overexpression of *GFP-rhi* was not able to rescue the reduced Rhi occupancy at 42AB upon *enok* knockdown (Fig 7C). We also asked whether the regulation of Rhi recruitment by Enok is dependent on its HAT activity. To answer this question, we examined whether overexpressing *rhi* can rescue the defective transcription at 42AB in the *enok²* mutant germline clone ovaries, as the *enok²* allele encodes a full-length Enok protein with compromised HAT function. Interestingly, overexpression of *GFP-rhi* failed to rescue the decreased levels of RNAs derived from 42AB in the *enok²* mutant (S15E and S15F Fig). These results strongly suggest that Enok plays a critical role, which is dependent on its HAT activity and cannot be bypassed by overexpressing *rhi*, in Rhi recruitment to some specific Enok-dependent piRNA source loci.

Taken together, our results suggest that Enok contributes to proper transposon silencing in the ovarian germline through three pathways summarized in Fig 7D. First, Enok acetylates H3K23 at genes involved in the piRNA production, mainly *rhi*, and promotes their expression (left). Second, Enok regulates the recruitment of Rhi to enhance transcription at piRNA clusters, facilitating the synthesis of piRNA precursors (middle). Third, as only 3 out of 7 activated transposon families (Fig 1C) showed decreased antisense piRNA levels in *enok* mutants (S8A Fig), Enok may regulate silencing of the other 4 transposons by unknown mechanisms (right).

## Discussion

We report herein a novel role for Enok in suppressing the activation of transposons in the germline. Loss of functional Enok in the ovarian germline resulted in activation of 7 transposon families (Fig 1C). This amount of activated transposon families in *enok* mutant ovaries is comparable to the 17 families activated in the *rhi* mutant [15]. Our RNA-seq analysis showed a ~75% reduction in the mRNA levels of *rhi* in *enok* mutant germline clone ovaries as compared with the WT control (Figs 3A and S7A). Knocking down *enok* in the ovarian germline using two different *UAS-shRNA-enok* constructs also reduced the mRNA levels of *rhi* as compared with two different control fly lines (Figs 4A and S10A). In addition, Enok ChIP-seq analysis revealed that Enok is localized to the 5' region of *rhi* (Fig 3B), and the Enok-dependent enrichment of H3K23ac at the 5' end of *rhi* suggests that the Enok-mediated H3K23ac mark promotes *rhi* expression, contributing to proper piRNA production (Fig 3D and 3E). Indeed, *enok* mutant germline clone ovaries showed decreased levels of piRNAs that mapped to RD-SL (Fig 2B). However, not all RD-SL showed decreased piRNA levels in *enok* mutants. About 20% of the 6426 RD-SL showed reduced piRNA levels in *enok* mutants (Fig 2C and S3 Table). Therefore, the remaining 25% of *rhi* levels in *enok* mutant ovaries (Fig 3A) may be sufficient to support transcription of the RD-SL that were not affected by loss of Enok. More strikingly, knocking down *enok* in the germline, without affecting the global protein levels of Rhi (S12B–S12E Fig), reduced Rhi occupancies at ED-SL but not at EI-SL (Fig 6C). This result suggests that Enok regulates Rhi recruitment specifically at ED-SL. The *enok* and *rhi* mutants show similar effects on the fold changes in transposon family expression and in antisense piRNAs (S8B Fig) [15]. However, among the top 24 most highly overexpressed families in *rhi* [15], loss of Enok in the germline specifically activates 7 families. This specificity suggests that these 7 families may be more sensitive to reductions in Rhi

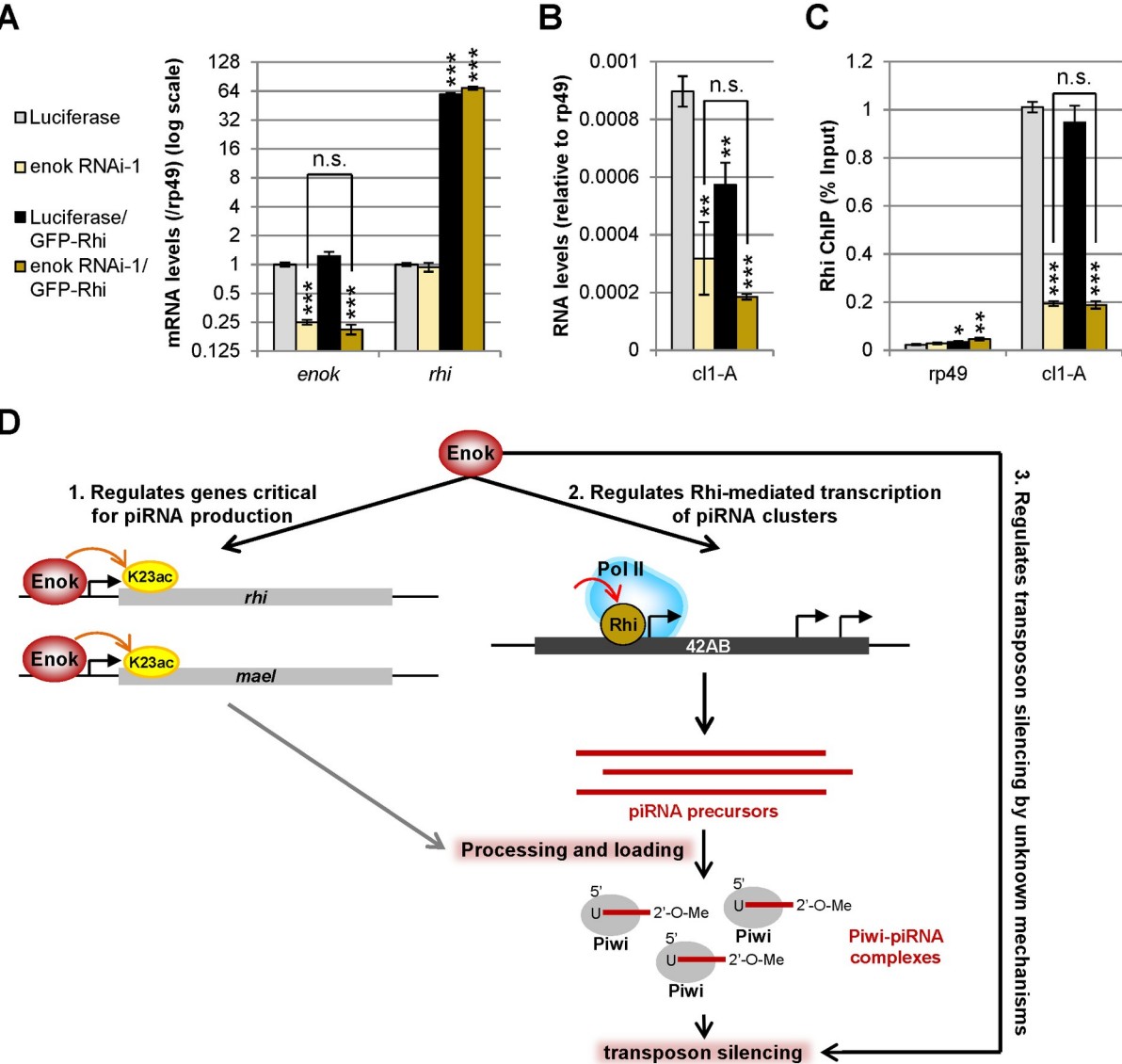

**Fig 7. Overexpressing *rhi* cannot rescue the defective Rhi occupancy at 42AB upon *enok* knockdown. (A)** The efficiency of *enok* knockdown was similar with or without *GFP-rhi* overexpression. RT-qPCR analysis of ovaries was used to examine expression levels of the indicated genes. The mRNA levels were normalized to the levels of *rp49*. **(B)** Overexpression of *GFP-rhi* cannot rescue the defective transcription at 42AB upon *enok* knockdown. The same total RNA samples used in (A) were subjected to strand-specific RT-qPCR analysis for RNAs derived from the dual-strand cluster 42AB. **(C)** Overexpression of *GFP-rhi* cannot rescue the defective Rhi recruitment to 42AB upon *enok* knockdown. The Rhi levels at 42AB in ovaries with or without overexpressed GFP-Rhi were analyzed by ChIP-qPCR. **(D)** Model of the regulation of piRNA production by Enok in the ovarian germline. Left panel: Enok is localized to and acetylates H3K23ac at genes involved in piRNA biogenesis to promote their expression. Middle panel: Enok facilitates transcription at a subset of piRNA clusters by regulating Rhi recruitment. Right panel: Enok may have other unknown roles in regulating transposon silencing. In (A-C), data represent the mean of three biological replicates +/- SD. The location of amplicons used for 42AB is the same as indicated in Fig 4B. *P < 0.05, **P < 0.01, ***P < 0.001 (Student's t-test). n.s.: not significant. Genotypes of the females are as follows: + / *CyO*; *P{w[+mC] = GAL4::VP16-nos.UTR} CG6325^{MVD1}* / *P{UAS-LUC.VALIUM10}attP2* (Luciferase); + / *CyO*; *P{w[+mC] = GAL4::VP16-nos.UTR}CG6325^{MVD1}* / *P{TRiP.HMS02634} attP2* (enok RNAi-1); + / *CyO*; *P{w[+mC] = GAL4::VP16-nos.UTR}CG6325^{MVD1}*, *UASp-rhi:GFP* / *P{UAS-LUC.VALIUM10}attP2* (Luciferase/ GFP-Rhi); +/*CyO*; *P{w[+mC] = GAL4::VP16-nos.UTR}CG6325^{MVD1}*, *UASp-rhi:GFP/P{TRiP.HMS02634}attP2* (enok RNAi-1/GFP-Rhi).

recruitment to a subset of piRNA source loci. Taken together, Enok may contribute to fine-tuning transcription of piRNA clusters by modulating *rhi* expression and by regulating Rhi recruitment to Enok-dependent piRNA source loci.

Three genome-wide RNAi screens have been reported before, but two of them were specifically performed in ovarian somatic cells [27,28]. Knocking down *enok* in ovarian somatic cells using the tj-Gal4 driver did not activate the soma-dominant transposon, *Gtwin* (S16A Fig), suggesting that Enok may be dispensable for transposon silencing in the soma. In the genome-wide screen in the germline [25], the *enok* RNAi construct (KK108400) is a long hairpin RNA. The efficiency of knocking down *enok* by long hairpin RNAs is lower than by short hairpin RNAs in the germline, even in the presence of additional Dicer-2 (S16B Fig). Czech et al. indeed showed that knocking down *enok* weakly activated the *blood* and *Burdock* transposons (z-scores of -0.5 and -0.74, respectively). However, this activation effect did not reach the threshold (z-score of -1.5 or lower) applied in the screen. We used two different short hairpin RNA constructs against *enok* to deplete Enok in the germline, and therefore we were able to detect stronger activation of transposons (Figs 4A and S10A), possibly due to better knockdown efficiencies.

Enok is the major enzyme responsible for the abundant H3K23ac mark [20,21]. It was previously demonstrated that Enok is localized to the 5' end of its target genes, *spir* and *mael*, and promotes their expression by acetylating H3K23 [20]. Here we further report *rhi* and a subset of RD-SL (defined as ED-SL) as novel targets that are transcriptionally regulated by Enok. Intriguingly, while the 5' region of *rhi* is enriched with Enok and H3K23ac (Fig 3B and 3D), Enok is not enriched at ED-SL relative to EI-SL (S13A Fig). Also, knocking down *enok* in ovaries reduced the H3K23ac levels at the 5' end of *rhi* (Fig 3E) but not at piRNA clusters (S6 Table). These results suggest that Enok facilitates *rhi* expression by acetylating H3K23, but regulates the transcription of ED-SL through other mechanisms. Notably, knocking down *enok* in the ovarian germline severely reduced the Rhi occupancy to sites in 42AB (Fig 6A and 6B), while global protein levels of Rhi and the H3K9me3 levels at 42AB were largely unaffected (S12B–S12E and S14C Figs). Therefore, Enok is likely to promote transcription of ED-SL by regulating Rhi recruitment.

The transcription of dual-strand piRNA clusters utilizes noncanonical heterochromatin-dependent internal initiation. Transcription initiation at these clusters was proposed to take place by the H3K9me3-bound RDC complex recruiting the germline-specific paralog of TFIIA-L, Moon [6]. In this study, we show that Enok is important for both Rhi and Pol II occupancies at a subset of RD-SL (defined as ED-SL) (Figs 5B and 6C), suggesting that Enok can facilitate transcription of these piRNA source loci. As Rhi is highly enriched across the entire 42AB cluster [14] and no Enok peaks were detected within 42AB (Fig 5A), Enok is unlikely to regulate the Rhi occupancy at 42AB by directly recruiting it. Also, our Co-IP assay failed to detect interaction between Enok and the overexpressed Rhi in ovaries. Interestingly, the HAT activity of Enok is critical for transcription of 42AB even when *rhi* is overexpressed (S15E and S15F Fig). Therefore, it is possible that Enok may play a role in acetylating some factors that are required for Rhi recruitment, or it may have an indirect role in Rhi recruitment by promoting expression of other genes with yet unidentified functions in the piRNA pathway. Notably, while knocking down *enok* in the germline decreased the RNA levels transcribed from both genomic strands at cl1-A and from the sense strand at cl1-32, RNA levels transcribed from the antisense strand at cl1-32 was not affected by depletion of Enok (Figs 4B and S10B). Thus, within dual-strand clusters, Enok may regulate the internal initiation in specific regions. Taken together, our study provides novel information regarding noncanonical transcription and transposon silencing in the germline.

## Materials and methods

### Fly strains and culture

*Drosophila melanogaster* was crossed and grown on standard media at 25°C unless stated otherwise. To generate germline clone ovaries, larvae were heat shocked twice at 37°C for 40 min,

with 30 min break at 25˚C in between, for two consecutive days starting from the second day after hatching. Females were conditioned with wet yeast for 2–3 days before ovary dissection. MTD-Gal4 driven RNAi and overexpression of transgenes were carried out at 29˚C for 2–3 days. Females were conditioned with wet yeast for 1–2 days before ovary dissection.

The following fly lines were obtained from the Bloomington *Drosophila* Stock Center with their stock numbers in parentheses: *UAS-Luciferase* (35788), *UAS-enok RNAi-1* (40917), *UAS-enok RNAi-2* (41664), *UAS-enok RNAi-3* (29518), *UAS-EGFP RNAi-1* (41556), *UAS-Dcr2; nos-GAL4* (25751) and *MTD-GAL4* (31777). The fly lines containing the *MTD-GAL4* driver and a *GFP-rhi* or *GFP-del* transgene were obtained from the JB-STOCK Library in the Vienna *Drosophila* Resource Center (stock number 313667 and 313672) [14]. The fly line containing the *nos-Gal4-VP16* driver and a *UASp-rhi:GFP* transgene was kindly provided by William E. Theurkauf [15].

## RNA purification and RT-qPCR

RNA isolation and cDNA synthesis were performed essentially as described previously with minor modifications [20]. Briefly, total RNA was isolated using TRIzol (Invitrogen) or REzol (Protech), treated with DNase (Invitrogen), and subjected to cDNA synthesis using SuperscriptIII reverse transcriptase (Invitrogen) or M-MuLV reverse transcriptase (Protech). For examination of the mRNA levels of transposons or genes, qPCR was performed in triplicate for each biological replicates, and the data were analyzed using the standard curve method. The mRNA levels of *rp49* were used as the normalization control. For the strand-specific RT-qPCR analysis for RNAs derived from piRNA clusters, qPCR data were analyzed by the ΔΔCt method, using the transcript levels of *rp49* in each sample as the internal normalization standard, which was set as 1. For each biological replicate, the mean of technical triplicates was used. The p-values were calculated using these means by T-tests

## RNA-seq analysis

RNA-seq data were published previously [20], and were retrieved from the ArrayExpress database (www.ebi.ac.uk/arrayexpress; accession number E-MTAB-2521). The RPKM values for each gene were calculated as described in the previous study [20]. The expression levels of transposon families were calculated with piPipes 1.5.0, using the default RNA-seq pipeline with UCSC dm3 as the reference genome (0 mismatches; multiple mappers assigned using Expectation-Maximization algorithm; normalized to genome-mapping reads) [29]. Reads mapping to the transposon sequences from flyBase were normalized to transposon length and library depth (total mapped reads) to obtain the RPKM values for each transposon family. Transposon families activated in *enok* mutant ovaries were identified by TEtranscripts [30] using the cutoff of adjusted *P*-value < 0.05, fold change > 1.5 and RPKM > 1 in the *enok¹* and *enok²* data sets. For S4 Table, only the unique mappers were used (library depth for RPKMs: uniquely mapped reads), and the RNA-seq data were analyzed by edgeR [31] using UCSC dm3 and dm6, respectively (cutoff: FDR < 0.05). For S5 Table, only the unique mappers were used, and reads mapping to piRNA clusters were analyzed by DESeq2 [32] using UCSC dm3 (cutoff: adjusted *P*-value < 0.05).

## Small RNA-seq analysis

Ovaries were dissected from 3–5 day old flies. Total RNA was isolated from egg chambers of stages 5–14 using TRIzol (Invitrogen) followed by DNase treatment (Invitrogen), and subjected to small RNA-seq analysis.

Small RNA libraries were generated using TruSeq Small RNA Sample Preparation Kit (RS-200-0012; Illumina) with the following modification. Size selection was done by purifying 18–30 nt long small RNAs from a polyacrylamide gel. A 2S blocking DNA oligo for *Drosophila* rRNA was added (at 65˚C; 5 min) between the step of 3' primer Stop solution incubation and the step of 5' primer incubation. The resulting libraries were further size-selected by excising 135–160 nt long libraries from a polyacrylamide gel before sequencing on a HiSeq2500 (Illumina).

Reads generated were 51-base-pair (bp) single-end, directional using the Illumina protocol. The abundance of piRNAs mapping to each piRNA cluster or genic region shown in S2 Table were calculated with piPipes 1.4.12, using the default small RNA-seq pipeline with UCSC dm3 as the reference genome (0 mismatches; only unique mappers; normalized to miRNAs) [29]. Mapping statistics are shown in S1 Fig. The piRNA clusters with decreased or increased levels of piRNAs mapping to them in *enok* mutant ovaries were identified by DESeq2 [32] using the cutoff of adjusted *P*-value < 0.05.

For the 1 kb bin analysis, only the unique mappers were used. The definition of RD-SL, RI-SL, SO-SL and het non-SL loci was obtained from the previous study [14]. After normalized to library depth (uniquely mapped reads), the mapped reads from 3 independent replicates were pooled and normalized to mappability [14], and bins with zero reads were excluded from the analysis. The Wilcoxon Rank sum test with continuity correction was used to calculate *P* values. As mentioned in the previous study [14], a *P* value of $<2.2 \times 10^{-16}$ is the maximum R (wilcox.test) can accurately calculate and therefore the *P* values less than $2.2 \times 10^{-16}$ were shown as "*P* < 2.2e-16".

## ChIP-seq analysis

Ovaries were dissected from 2–3 day old flies in Grace's insect medium. For Enok ChIP and Pol II ChIP, 360 and 120 pairs of ovaries were used for each replicate, respectively. For Rhi and H3/H3K23ac ChIP, 90 and 60 pairs of ovaries were used for each replicate, respectively. ChIP assay was carried out as described previously with minor modifications [20]. Briefly, after fixation and homogenization, the chromatin pellet was sonicated in Buffer A2 (15 mM HEPES pH 7.5, 140 mM NaCl, 1 mM EDTA, 0.5 mM EGTA, 1% TritonX-100, 0.1% sodium deoxycholate, 0.1% SDS, 0.5% N-lauroyl sarcosine, 1 mM PMSF, 5 mM NaF, 5 mM sodium butyrate, 1 μg/mL pepstatin A, 1 μg/mL leupeptin and 1 μg/mL aprotinin) for 7 x 9 sec at 30% power. Following centrifugation, clarified soluble chromatin was incubated with α-Enok (13 μl) [20], α-RNA Pol II CTD (ab5408; Abcam; 2.5 μl), α-H3K23ac (07–355; Millipore; 2 μl), α-H3 (ab1791; Abcam; 2 μl) or α-Rhi (gift from Julius Brennecke; 3 μl) [14] overnight at 4˚C. After wash and elution, samples were treated with RNase A (QIAGEN) and proteinase K (Worthington). Crosslinking was then reversed at 65˚C for 6 hr, and DNA was purified by phenol-chloroform extraction and ethanol precipitation.

Reads generated were 101 bp single-end using the Illumina protocol. Fastq files were aligned to UCSC dm3 with Bowtie2-2.2.1 (0 mismatches; only unique mappers) [33]. Mapping statistics are shown in S1 Fig. Bam files were converted to bigwig files with bam2wig, additional parameters "-m" (https://github.com/MikeAxtell/bam2wig), and with wigToBigWig (UCSC). The bigwig files of $\log_2(FC)(IP/IN)(enok\_KD/GFP\_KD)$ were generated with the bigwigCompare tool from deepTools as follows [34]. Bigwig files of $\log_2(FC)(enok\_KD/GFP\_KD)$ were first generated for IP and input separately with additional parameters "—pseudocount 0.1—ratio log2". Then, bigwig files of $\log_2(FC)(IP/IN)(enok\_KD/GFP\_KD)$ were generated using the bigwig file of $\log_2(FC)(enok\_KD/GFP\_KD)$ for IP as "-b1" and that of $\log_2(FC)(enok\_KD/GFP\_KD)$ for input as "-b2", additional parameters "—pseudocount 0.01—ratio subtract".

For the 1 kb bin analysis, only the unique mappers were used, and the definition of RD-SL, RI-SL, SO-SL and het non-SL loci was obtained from the previous study [14]. After normalized to library depth (uniquely mapped reads), the mapped reads from independent replicates were pooled and normalized to mappability [14], and bins with zero reads were excluded from the analysis. The Wilcoxon Rank sum test with continuity correction was used to calculate $P$ values, and the $P$ values less than $2.2 \times 10^{-16}$ were shown as "$P < 2.2e\text{-}16$".

### Correlation analysis of the NGS data

The Spearman's correlation coefficients between each replicate/sample in our RNA-seq, small RNA-seq and ChIP-seq data sets were calculated with the plotCorrelation tool from deepTools using multiBamSummary, 10 kilobases by default [34], additional parameters "—skipZeros". The Principal Component Analysis (PCA) plot was generated using R.

### ChIP-qPCR analysis

Sixty pairs of ovaries were dissected from 2–3 day old females in Grace's insect medium, and ChIP experiments were performed as described above in the ChIP-seq Analysis section. The following antibodies were used: α-Enok (guinea pig; 4 μl) [20], α-H3K9me3 (39161; Active Motif; 3 μl), α-H3K23ac (07–355; Millipore; 2 μl), α-H3 (ab1791; Abcam; 2 μl) or α-Rhi (gift from Julius Brennecke; 2 μl) [14]. The qPCR data obtained from input and immunoprecipitated (IP) DNA samples were analyzed using the standard curve method.

### Immunofluorescent staining

Immunofluorescent staining of ovaries was carried out as previously described [20]. Ovaries were dissected in Grace's insect medium. The following primary and secondary antibodies were used: α-Piwi 1:200 (sc-98264; Santa Cruz), α-Rhi 1:200 (S7C Fig) or 1:500 (S12B Fig) (gift from Julius Brennecke) [14], α-Fib 1:100 (MCA-38F3; EnCor Biotechnology), α-Mael 1:50 (gift from Haruhiko Siomi) [35], α-H3K23ac 1:2500 (07–355; Millipore), Alexa 488 goat anti-mouse 1:400 (Molecular Probes) and Cy3 donkey anti-rabbit 1:400 (Jackson ImmunoResearch). Cell nuclei were stained using DAPI dye (Invitrogen).

### Western blot analysis

Ten pairs of ovaries were dissected in Grace's insect medium. Whole cell extracts (WCEs) were prepared by lysing ovaries in 1X SDS sample buffer (62.5 mM Tris-HCl pH6.8, 10% glycerol, 2% SDS, 0.01% bromophenol blue, and 143 mM β-mercaptoethanol). After heated for 6 min at 98°C, lysates were centrifuged at 20000g for 5 min and supernatants were collected as WCEs. The following primary antibodies were used in western blot analysis: GFP 1:1000 (11814460001; Roche), GFP 1:1000 (632381; Clontech), H3 1:10000 (ab1791; Abcam), and β-Tubulin 1:5000 (E7; DSHB).

### Primers

Primers used in strand-specific RT reaction are as follows:

cl1-A-plus, 5'-CGAAGCCTTAGATCTCGCTCC-3';

cl1-A-minus, 5'-ACATCAGGAACACAGCGAGGTG-3';

cl1-32-plus, 5'-GGTGCAAATGTCTCATCATAATCAGTC-3';

cl1-32-minus, 5'-GATGAAATTGAATTTCGTGATGACAGATC-3';

cl2-A-plus, 5'-CGTGGGTCCAG-3';

cl2-A-minus, 5'-TGAAAATCGCATC-3';

*rp49*-rt, 5'-GGAGGAGACGCCG-3'.

   Primers used in qPCR are as follows:

*rp49* F, 5'-GACGCTTCAAGGGACAGTATCTG-3';

*rp49* R, 5'-AAACGCGGTTCTGCATGAG-3';

*enok* F, 5'-TGTTTTTGTGCGAGTTCTGC-3';

*enok* R, 5'-AGTAGCCAACCAAATGGCAC-3';

*enok* F (S16B Fig), 5'-TCAAAGTCAAGCGGATACTG-3';

*enok* R (S16B Fig), 5'-CCGTTTTTGCCACTTTAACC-3';

*Burdock* F, 5'-CGGTAAAATCGCTTCATGGT-3';

*Burdock* R, 5'-ACGTTGCATTTCCCTGTTTC-3';

*Blood* F, 5'-TGCCACAGTACCTGATTTCG-3';

*Blood* R, 5'-GATTCGCCTTTTACGTTTGC-3';

*rhi* F, 5'-GAAACCCTTCGGTGTGAATC-3';

*rhi* R, 5'-ACTTGGGCACAATGATCCTC-3';

*mael* F, 5'-CCACGAGGATAATGATCGAACCAAG-3';

*mael* R, 5'-GCGAATAAAAAGAGTCGAAGGAGGA-3';

*BoYb* F, 5'-GTACAGGAGTTCCGCATGCA-3';

*BoYb* R, 5'-CTGGAGTTTCACTGTCAAATC-3';

*cuff* F, 5'-CGTTCGTTTCGGACTGGTTCC-3';

*cuff* R, 5'-GACATGGTTTGCAAATTCGGTG-3';

*del* F, 5'-CAGCTACATGCATAATGAAAGTGC-3';

*del* R, 5'-GAATAATGAAAGCTAGGGTTTGGC-3';

cl1-A F, 5'-CGTCCCAGCCTACCTAGTCA-3';

cl1-A R, 5'-ACTTCCCGGTGAAGACTCCT-3';

cl1-32 F, 5'-GTGGAGTTTGGTGCAGAAGC-3';

cl1-32 R, 5'-AGCCGTGCTTTATGCTTTAC-3';

cl2-A F, 5'-GCCTACGCAGAGGCCTAAGT-3';

cl2-A R, 5'-CAGATGTGGTCCAGTTGTGC-3';

*mael* 5' F, 5'-ATAGATATTTGAAGTGTCAGCGGAATCGAC-3';

*mael* 5' R, 5'-GAAATGGTTACTTCGGAATGTCGCG-3';

*rhi* 5' F, 5'-GCTTTCCCATCCCTAATAGTATTCCG-3';

*rhi* 5' R, 5'-TCATTAGGCGGTGCATCGACCAG-3';

*shu* 5' F, 5'-ACGAATAGGACAAGGGATTCTTGAGC-3';

*shu* 5' R, 5'-GAAAGCTCTTTTCTCAAACGCATCC-3';

intergenic region F, 5'-GAGCAGACAACGCTCCAAGACCCAA-3';

intergenic region R, 5'-AAATTTTCCACCTACCTGCCGCACG-3';

*Burdock* 5' F, 5'-ATTAGAAGCGTCGGTCATCG-3';

*Burdock* 5' R, 5'-GGGCGCCAATTATCATTTTA-3';

*HeT-A* pro F, 5'-ACCACGCCCAACCCCCAA-3';

*HeT-A* pro R, 5'-GCTGGTGGAGGTACGGAGACAG-3';

*Gtwin* F, 5'-TTCGCACAAGCGATGATAAG-3';

*Gtwin* R, 5'-GATTGTTGTACGGCGACCTT-3'.

All amplicons for *Burdock*, *Blood* or *HeT-A* are not located within 42AB.

## Supporting information

**S1 Fig. Mapping statistics of the small RNA-seq and ChIP-seq data sets.**
(TIFF)

**S2 Fig. Replicates in the RNA-seq, small RNA-seq and ChIP-seq analyses are highly correlated.** Spearman's correlation coefficients across different samples and replicates in the RNA-seq (A), small RNA-seq (B), Enok ChIP-seq (C), RNA Pol II ChIP-seq (D) and H3K23ac/H3 ChIP-seq (E) analyses were calculated using plotCorrelation from deepTools. The color scale used for all tables is shown in the bottom panel of (C).
(TIFF)

**S3 Fig. Principal Component Analysis (PCA) plot of RNA-seq data sets.** The PCA plot of RNA-seq samples was generated using R.
(TIFF)

**S4 Fig. Small RNA-seq analysis using unique mappers showed that 54% of the piRNA clusters were down-regulated in the two *enok* mutants. (A)** Venn diagrams demonstrating the overlap between the piRNA clusters with different piRNA levels identified by small RNA-seq in the *enok¹* and *enok²* ovaries compared with the WT control. Differential analysis was performed by DESeq2 using the cutoff of adjusted p-value < 0.05. Clusters with down-regulated or up-regulated piRNA levels in mutants compared with the WT control are shown in the top or the bottom panel, respectively. **(B)** Genome Browser view of small RNA-seq results of four examples of piRNA clusters that either were down-regulated (42AB and cluster 29; left panel), remained unaffected (cluster 2; right top panel) or up-regulated (cluster 73; right bottom panel) in *enok* mutants as compared with the WT control. In (A-B), genotypes of females are as follows: *hs-Flp / +; FRT^G13 / FRT^G13, ovo^D1-18* (WT); *hs-Flp / +; FRT^G13, enok¹/ FRT^G13, ovo^D1-18* (*enok¹*); *hs-Flp/+; FRT^G13, enok²/FRT^G13, ovo^D1-18* (*enok²*).
(TIFF)

**S5 Fig. piRNA production from 42AB is severely reduced in *enok* mutants. (A)** Upper panel: Distribution of overlap sizes between pairs of complementary piRNAs uniquely mapping to 42AB with ping-pong z-scores [36] in the WT and *enok* mutant germline clone ovaries.

Reads of complementary piRNA pairs were normalized to miRNA. Lower panel: Length histograms of piRNAs uniquely mapping to 42AB in the WT and *enok* mutant germline clone ovaries. Reads were normalized to miRNA. **(B)** Length histograms of piRNAs mapping to transposons in the WT and *enok* mutant germline clone ovaries. **(C)** Sense and antisense piRNA reads mapping to the consensus *Burdock* sequence. In (A-C), genotypes are as described in S4 Fig.
(TIFF)

**S6 Fig. Characterization of Enok-dependent and -independent piRNA source loci. (A)** Scatter-plot displaying piRNA fold changes in *enok*[1] (x axis) versus *enok*[2] (y axis) mutants for all Rhi-dependent piRNA source loci (RD-SL). The two classes (ED-SL and EI-SL) and their respective population sizes are indicated. The color gradient (yellow < red < black) indicates the density of underlying 1 kb bins. RD-SL in 42AB, cluster5 or cluster38C1 are indicated using blue, green or grey dots, respectively. **(B-C)** Venn diagrams demonstrating the overlap between the RD-SL with different piRNA levels identified by small RNA-seq in the *enok*[1] and *enok*[2] ovaries compared with the WT control. RD-SLs with down-regulated or up-regulated piRNA levels in mutants compared with the WT control are shown in (B) and (C), respectively. In (A-C), genotypes are as described in S4 Fig.
(TIFF)

**S7 Fig. Five genes involved in piRNA biosynthesis were commonly down-regulated in the two *enok* mutants as compared with the WT control, while the expression levels of *piwi*, *AGO3* and *aub* were not decreased in *enok* mutants. (A)** The expression levels of selected genes involved in piRNA biosynthesis in the WT control and *enok* mutant ovaries are shown in RPKM obtained from RNA-seq (dm3). Data represent the mean of three biological replicates +/- SD. *FDR < 0.05, **FDR < 0.01, ***FDR < 0.001 (edgeR). **(B)** Stage 10 egg chambers were stained with DAPI and an α-Piwi antibody. Bars: 50μm. **(C)** Stage 8 egg chambers were stained with α-Rhi and α-Fibrillarin (Fib) antibodies. Projections of 16 sections in the middle of egg chambers are shown. Fib staining is shown as a staining control. Bars: 25μm. **(D)** Stage 10 egg chambers were stained with DAPI and an α-Mael antibody. Bars: 50μm. Genotypes in (A-D) are as described in S4 Fig.
(TIFF)

**S8 Fig. The *enok* mutants showed effects on transposon expression and piRNA abundance similar to the *rhi* mutant. (A)** Reads of antisense piRNAs encoded by transposon families (normalized to miRNAs) in two *enok* mutants are plotted against those in the WT control. The seven transposon families that are significantly activated in *enok* mutants are labeled by red dots. **(B)** Fold changes in transposon family expression in two *enok* mutants (y axis, mutant/WT) plotted against fold changes in antisense piRNAs encoded by the same transposon family (x axis, mutant/WT) are shown. The highly over-expressed transposons (*Burdock* and *HeT-A*) also showed large reductions in antisense piRNAs. However, expression of *blood* increased 10–12 fold while the total antisense piRNA pool was only reduced by 12%-25%. In (A-B), genotypes are as described in S4 Fig.
(TIFF)

**S9 Fig. Genome Browser view of Enok ChIP-seq data using two α-Enok antibodies (#1 and #2) raised independently in two guinea pigs. (A)** Enok peaks across a 200 kb region, including the *mael* gene, were reproducibly detected in 3 independent replicates of ChIP-seq analysis. **(B)** Genome Browser view of Enok ChIP-seq data at *CG4239* and *elg1*. *CG4239* represents a negative control with no Enok enrichment and no changes in expression in *enok* mutants, and *elg1* represents a non-piRNA pathway germline-specific gene enriched by Enok.

Experimental details are as described in the legend for Fig 3B.
(TIFF)

**S10 Fig. Knocking down *enok* in the germline resulted in down-regulation of *rhi* and transcript levels from 42AB. (A)** RT-qPCR analysis of ovaries was used to examine the expression levels of the indicated transposon and genes. The mRNA levels were normalized to the levels of *rp49*. **(B)** The same total RNA samples used in (A) were subjected to strand-specific RT-qPCR analysis for RNAs derived from 42AB (left panel) and cluster 2 (right panel). The bottom panel shows the signals of piRNA reads mapping to the indicated clusters in WT ovaries as described in Fig 2. Red bars in the bottom panel indicate the location of amplicons used in the qPCR reaction. In (A-B), data represent the mean of three biological replicates +/- SD. *P < 0.05, **P < 0.01, ***P < 0.001 (Student's t-test). Genotypes of the females are as follows: *P{w[+mC] = otu-GAL4::VP16.R}1 / +; P{w[+mC] = GAL4-nos.NGT}40 / +; P{w[+mC] = GAL4::VP16-nos.UTR}CG6325$^{MVD1}$ / P{VALIUM20-EGFP.shRNA.1}attP2* (EGFP RNAi-1); *P {w[+mC] = otu-GAL4::VP16.R}1 / +; P{w[+mC] = GAL4-nos.NGT}40 / +; P{w[+mC] = GAL4:: VP16-nos.UTR}CG6325$^{MVD1}$ / P{TRiP.HMS02634}attP2* (enok RNAi-1); *P{w[+mC] = otu-GAL4::VP16.R}1 / +; P{w[+mC] = GAL4-nos.NGT}40/+; P{w[+mC] = GAL4::VP16-nos.UTR} CG6325$^{MVD1}$/P{TRiP.HMS02048}attP2* (enok RNAi-2).
(TIFF)

**S11 Fig. Enok is important for levels of RNAs derived from a subset of piRNA clusters.** The levels of poly-A selected RNAs that uniquely mapped to piRNA clusters in *enok* mutant germline clone ovaries versus the WT control are plotted in a log scale. Selected piRNA clusters that were significantly down-regulated in *enok* mutants are indicated by red dots. Genotypes are as described in S4 Fig.
(TIFF)

**S12 Fig. The protein levels of Rhi and Mael were largely unaffected in the *enok* knockdown ovaries. (A)** Stage 10 egg chambers were stained with DAPI and an α-Mael antibody. Bars: 50μm. **(B)** Stage 7 egg chambers were stained with DAPI and an α-Rhi antibody. Left panel: projections of 32 sections in the middle of egg chambers are shown. Right panel: enlarged images of the entire nuclei indicated by white solid boxes in the left panel are shown. Bars: 10μm. **(C)** Whole cell extracts were prepared from ovaries and subjected to western blotting. H3 was used as the loading control. The EGFP RNAi-1 knockdown, which targets the GFP-tag of GFP-Rhi and GFP-Del, was used as a negative control. Asterisks indicate non-specific bands detected by the α-GFP antibody. **(D-E)** Stage 8 egg chambers were stained with DAPI and an α-H3K23ac antibody. Representative images are shown in (D). Bars: 20μm. The staining signals of H3K23ac or the GFP signals in the nucleus of nurse cells were quantified and normalized to the DAPI signals. The normalized quantitation results are shown in (E). Data represent the mean +/- SD. All *P*-values between Luciferase+GFP-Rhi and other ovaries less than 0.001 are indicated using three (***) asterisks (Student's t-test). n.s.: not significant. In (A-B), genotypes are as described in S10 Fig. In (C-E), genotypes are as follows: *P{w[+mC] = otu-GAL4::VP16.R}1 / +; P{w[+mC] = GAL4-nos.NGT}40 / +; P{w[+mC] = GAL4::VP16-nos. UTR}CG6325$^{MVD1}$, P{rhi-GFP} / P{VALIUM20-EGFP.shRNA.1}attP2* (EGFP RNAi-1+-GFP-Rhi); *P{w[+mC] = otu-GAL4::VP16.R}1 / +; P{w[+mC] = GAL4-nos.NGT}40 / +; P{w [+mC] = GAL4::VP16-nos.UTR}CG6325$^{MVD1}$, P{rhi-GFP} / P{UAS-LUC.VALIUM10}attP2* (Luciferase+GFP-Rhi); *P{w[+mC] = otu-GAL4::VP16.R}1 / +; P{w[+mC] = GAL4-nos.NGT}40 / +; P{w[+mC] = GAL4::VP16-nos.UTR}CG6325$^{MVD1}$, P{rhi-GFP} / P{TRiP.HMS02634}attP2* (enok RNAi-1+GFP-Rhi); *P{w[+mC] = otu-GAL4::VP16.R}1 / +; P{w[+mC] = GAL4-nos. NGT}40 / +; TM3,Ser/ P{UAS-LUC.VALIUM10}attP2* (Luciferase; no GFP-Rhi); *P{w[+mC] =*

*otu-GAL4::VP16.R}1 / +; P{w[+mC] = GAL4-nos.NGT}40 / +; P{w[+mC] = GAL4::VP16-nos.UTR}CG6325^{MVD1}, P{del-GFP} / P{VALIUM20-EGFP.shRNA.1}attP2* (EGFP RNAi-1+-GFP-Del); *P{w[+mC] = otu-GAL4::VP16.R}1 / +; P{w[+mC] = GAL4-nos.NGT}40 / +; P{w[+mC] = GAL4::VP16-nos.UTR}CG6325^{MVD1}, P{del-GFP} / P{UAS-LUC.VALIUM10}attP2* (Luciferase+GFP-Del); *P{w[+mC] = otu-GAL4::VP16.R}1/ +; P{w[+mC] = GAL4-nos.NGT}40/ +; P{w[+mC] = GAL4::VP16-nos.UTR}CG6325^{MVD1}, P{del-GFP} /P{TRiP.HMS02634}attP2* (enok RNAi-1+GFP-Del).
(TIFF)

**S13 Fig. Enok is not enriched at ED-SL and knocking down *enok* did not affect the H3K23ac levels at 42AB. (A)** A box-plot displaying the distribution of Enok occupancies for all 1 kb bins belonging to the indicated groups. Center line, median; box limits, upper and lower quartiles; whiskers, the 5th and 95th percentile (outliers not shown). *P*-values were calculated using Wilcoxon Rank sum test. n.s.: not significant. **(B)** Genome Browser view of H3K23ac and H3 ChIP-seq data at 42AB. ChIP-seq experiments were performed in two independent replicates, and results from one representative replicate are shown. Input (IN) and immunoprecipitation (IP) tracks are shown. Genotypes of the females are as follows: *tj-GAL4/+; P{w[+mC] = GAL4::VP16-nos.UTR}CG6325^{MVD1} / P{VALIUM20-EGFP.shRNA.1}attP2* (GFP_KD); *tj-GAL4/+; P{w[+mC] = GAL4::VP16-nos.UTR}CG6325^{MVD1}/P{TRiP.HMS02634}attP2* (*enok*_KD).
(TIFF)

**S14 Fig. Knocking down *enok* mildly reduced the H3K9me3 levels at 42AB. (A)** RT-qPCR analysis of ovaries was used to examine the expression levels of the indicated transposon and genes. The mRNA levels were normalized to the levels of *rp49*. **(B)** The same total RNA samples used in (A) were subjected to strand-specific RT-qPCR analysis for RNAs derived from 42AB. The location of amplicon used in the qPCR reaction is as indicated in S10B Fig. **(C)** The H3K9me3 levels at 42AB (cl1-A and cl1-32), the 5' region of *Burdock* (Burdock 5') and the promoter of *HeT-A* (HeT-A pro) in ovaries were analyzed by ChIP-qPCR. The H3K9me3 IP signals were first normalized to the histone H3 IP signals (H3K9me3/H3), and then the H3K9me3/H3 values were normalized to the mean of H3K9me3/H3 values obtained for the *rp49* loci in EGFP RNAi-1 ovaries, which was set as 1. The location of amplicons used for 42AB is the same as indicated in S10B Fig. The amplicon for Burdock 5' is located at the position 152–283 of a full-length *Burdock* insertion, and that for HeT-A pro is located at the position 4094–4240 of the *HeT-A{}4795* insertion [37]. In (A-C), data represent the mean of three biological replicates +/- SD. *P < 0.05, **P < 0.01, ***P < 0.001 (Student's t-test). Genotypes of the females are as described in S10 Fig.
(TIFF)

**S15 Fig. Overexpression of *GFP-rhi* could not rescue the defective transcription of 42AB in the *enok* knockdown ovaries. (A)** Whole cell extracts were prepared from ovaries and subjected to western blotting. β-tubulin was used as the loading control. **(B)** Ovarioles containing the germarium and egg chambers up to stage 9 were stained with DAPI. Bars: 50μm. **(C)** RT-qPCR analysis of ovaries was used to examine the expression levels of the indicated genes. The mRNA levels were normalized to the levels of *rp49*. **(D)** The same total RNA samples used in (C) were subjected to strand-specific RT-qPCR analysis for RNAs derived from 42AB. The location of amplicon used in the qPCR reaction is as indicated in S10B Fig. **(E)** RT-qPCR analysis of ovaries was used to examine the expression levels of the indicated gene or transposon. The mRNA levels were normalized to the levels of *rp49*. **(F)** The same total RNA samples used in (E) were subjected to strand-specific RT-qPCR analysis for RNAs derived from 42AB. The

location of amplicon used in the qPCR reaction is as indicated in S10B Fig. In (C-F), data represent the mean of three biological replicates +/- SD except for *enok$^2$*, which represent two biological replicates. *P < 0.05, **P < 0.01, ***P < 0.001 (Student's t-test). n.s.: not significant. Genotypes of the females are as follows: + / CyO; P{w[+mC] = GAL4::VP16-nos.UTR} CG6325$^{MVD1}$ / P{VALIUM20-EGFP.shRNA.1}attP2 (EGFP RNAi-1); + / CyO; P{w[+mC] = GAL4::VP16-nos.UTR}CG6325$^{MVD1}$/ P{TRiP.HMS02634}attP2 (enok RNAi-1); +/CyO; P{w [+mC] = GAL4::VP16-nos.UTR}CG6325$^{MVD1}$, UASp-rhi:GFP /P{TRiP.HMS02634}attP2 (enok RNAi-1/GFP-Rhi); hs-Flp / +; FRT$^{G13}$ / FRT$^{G13}$, ovo$^{D1-18}$ (WT); hs-Flp / +; FRT$^{G13}$/ FRT$^{G13}$, ovo$^{D1-18}$; P{w[+mC] = GAL4::VP16-nos.UTR}CG6325$^{MVD1}$, UASp-rhi:GFP / + (WT/GFP-Rhi); hs-Flp / +; FRT$^{G13}$, enok$^2$ / FRT$^{G13}$, ovo$^{D1-18}$ (enok$^2$); hs-Flp/ +; FRT$^{G13}$, enok$^2$/FRT$^{G13}$, ovo$^{D1-18}$; P{w[+mC] = GAL4::VP16-nos.UTR}CG6325$^{MVD1}$, UASp-rhi:GFP/+ (enok$^2$/GFP-Rhi). (TIFF)

**S16 Fig. The efficiency of knocking down *enok* by long hairpin RNAs is lower than by short hairpin RNAs. (A)** RT-qPCR analysis of ovaries was used to examine the expression levels of *Gtwin*. The mRNA levels were normalized to the levels of *rp49*. Genotypes of the females are as follows: *tj-GAL4/+; P{VALIUM20-EGFP.shRNA.1}attP2/+* (EGFP RNAi-1); *tj-GAL4/+; P{TRiP.HMS02634}attP2/+* (enok RNAi-1); *tj-GAL4/+; P{TRiP.HMS02048}attP2/+* (enok RNAi-2). **(B)** RT-qPCR analysis of ovaries was used to examine the expression levels of *enok*. The mRNA levels were normalized to the levels of *rp49*. Genotypes of the females are as follows: *P{w[+mC] = UAS-Dcr-2.D}1 / +; P{w[+mC] = GAL4-nos.NGT}40 / +; P{VALIUM20-EGFP.shRNA.1}attP2 / +* (EGFP RNAi-1); *P{w[+mC] = UAS-Dcr-2.D}1 / +; P{w[+mC] = GAL4-nos.NGT}40 / +; P{TRiP.HMS02634}attP2 / +* (enok RNAi-1); *P{w[+mC] = UAS-Dcr-2.D}1/ +; P{w[+mC] = GAL4-nos.NGT}40/ +; P{TRiP.HM05195}attP2/+* (enok RNAi-3). In (A-B), Data represent the mean of three biological replicates +/- SD except for EGFP RNAi-1 in (B), which represent two biological replicates. ***P < 0.001 (Student's t-test). (TIFF)

**S1 Table. Reads per kilobase per million (RPKMs) of transposons.** The RPKM values of each transposon family obtained from the RNA-seq analysis of triplicate samples of the WT ovary or the *enok* mutants were calculated using piPipes. Differential analysis was performed using TEtranscripts. Genotypes are as follows: *hs-Flp / +; FRT$^{G13}$ / FRT$^{G13}$, ovo$^{D1-18}$* (WT); *hs-Flp / +; FRT$^{G13}$, enok$^1$/ FRT$^{G13}$, ovo$^{D1-18}$* (enok1); *hs-Flp/+; FRT$^{G13}$, enok$^2$/FRT$^{G13}$, ovo$^{D1-18}$* (enok2). (XLSX)

**S2 Table. Abundance of piRNAs mapping to piRNA clusters and genic regions.** The abundance of piRNAs mapping to the sense or antisense strand of each piRNA clusters or genic regions obtained from the small RNA-seq analysis of triplicate samples of the WT ovary or the *enok* mutants are shown. The unique reads mapping to each piRNA cluster or genic region were normalized to miRNAs and defined as the abundance of piRNAs. The results of DESeq2 analysis on piRNA reads mapping to piRNA clusters are also shown. Genotypes are as described in S1 Table. (XLSX)

**S3 Table. Abundance of piRNAs mapping to ED-SL and EI-SL.** The abundance of piRNAs mapping to ED-SL or EI-SL obtained from the small RNA-seq analysis of triplicate samples of the WT ovary or the *enok* mutants are shown. The unique reads mapping to each ED-SL or EI-SL were normalized to library depth and mappability, and defined as the abundance of piRNAs. The location, mappability and transposon content of each ED-SL or EI-SL are also

shown. Genotypes are as described in S1 Table.
(XLSX)

**S4 Table. RPKMs of genes involved in the piRNA pathway.** The list of genes involved in the piRNA pathway was obtained from the previous transcriptome-wide RNAi screen [25]. The RPKM values obtained from the RNA-seq analysis of triplicate samples of the WT ovary or the *enok* mutants are shown (using UCSC dm3 or dm6 as the reference genome). Differential analysis was performed using edgeR. Genotypes are as described in S1 Table.
(XLSX)

**S5 Table. RNA levels at piRNA clusters.** Unique mappers from the RNA-seq analysis of triplicate samples of the WT ovary or the *enok* mutants mapping to each piRNA cluster were analyzed using DESeq2. The six clusters that were significantly down-regulated (adjusted p-value < 0.05) in the two *enok* mutants are highlighted in purple. Genotypes are as described in S1 Table.
(XLSX)

**S6 Table. H3K23ac levels at piRNA clusters and transposons.** Unique mappers from the H3K23ac and H3 ChIP-seq analyses of two replicates of the control (GFP_KD) or the *enok* knockdown (*enok*_KD) ovaries mapping to each piRNA cluster or transposon were analyzed using DESeq2. The H3 ChIP-seq data were used as the normalization control for the H3K23ac ChIP-seq data. The results of differential analysis of H3K23ac levels in *enok*_KD compared with GFP_KD are shown. Transposons or piRNA clusters with decreased H3K23ac levels upon *enok* knockdown were defined as log2FoldChange < -1 and adjusted p-value < 0.05. Genotypes are as follows: *tj-GAL4/+; P{w[+mC] = GAL4::VP16-nos.UTR}CG6325$^{MVD1}$ / P{VALIUM20-EGFP.shRNA.1}attP2* (GFP_KD); *tj-GAL4/+; P{w[+mC] = GAL4::VP16-nos.UTR}CG6325$^{MVD1}$/P{TRiP.HMS02634}attP2* (*enok*_KD).
(XLSX)

**S7 Table. Additional data used in box plots or bar charts.** Data used in box plots or bar charts that are not presented in S1–S6 Tables are shown.
(XLSX)

## Acknowledgments

We thank Bloomington Stock Center, VDRC and W.E. Theurkauf for fly stocks. We also thank the Developmental Studies Hybridoma Bank, J. Brennecke and H. Siomi for antibodies. We thank the High Throughput Sequencing Core, funded by Academia Sinica Core Facility and Innovative Instrument Project (AS-CFII-108-114), for acquiring the ChIP-seq data.

## Author Contributions

**Conceptualization:** Fu Huang.

**Data curation:** Shih-Ying Tsai, Fu Huang.

**Formal analysis:** Shih-Ying Tsai, Fu Huang.

**Funding acquisition:** Fu Huang.

**Investigation:** Shih-Ying Tsai, Fu Huang.

**Methodology:** Fu Huang.

**Project administration:** Fu Huang.

**Supervision:** Fu Huang.

**Validation:** Shih-Ying Tsai, Fu Huang.

**Visualization:** Shih-Ying Tsai, Fu Huang.

**Writing – original draft:** Fu Huang.

**Writing – review & editing:** Shih-Ying Tsai, Fu Huang.

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
