## [Decision Letter · Decision Letter 0]

19 Feb 2020

Dear Dr FU HUANG

Thank you very much for submitting your Research Article entitled 'Acetyltransferase Enok regulates transposon silencing by promoting piRNA cluster transcription' to PLOS Genetics. Your manuscript was fully evaluated at the editorial level and by independent peer reviewers. The reviewers appreciated the attention to an important problem, but raised some substantial concerns about the current manuscript. Based on the reviews, we will not be able to accept this version of the manuscript, but we would be willing to review again a much-revised version. We cannot, of course, promise publication at that time.

If you decide to revise the manuscript for further consideration at PLOS Genetics, please aim to resubmit within the next 60 days, unless it will take extra time to address the concerns of the reviewers, in which case we would appreciate an expected resubmission date by email to plosgenetics@plos.org.

[LINK]

We are sorry that we cannot be more positive about your manuscript at this stage. Please do not hesitate to contact us if you have any concerns or questions.

Yours sincerely,

Severine Chambeyron

Guest Editor

PLOS Genetics

Wendy Bickmore

Section Editor: Epigenetics

PLOS Genetics

Reviewer's Responses to Questions

**Comments to the Authors:**

Reviewer #1: Tsai and Huang describe a novel function of the Drosophila acetyltransferase Enok in transcription of piRNA precursors and Rhino recruitment to a subset of the germline piRNA clusters. The data presented in the manuscript undoubtedly have a potential interest for the piRNA field, however, the analysis seems to be superficial and not conclusive.

Specific comments

1. Lines 78-81. Information about the presumed function of H3K23ac histone mark in the Introduction is poor and should be enhanced.

2. Fig1A – can you define the Y axis (TPM, log2TPM?)

3. Figs S1, S2 – PCA plots are more appropriate to demonstrate similarity of RNAseq replicates

4. Different effects on TAHRE, hobo and rover expression in enok1 and enok2 could be explained by the inter-strain polymorphism in the TE copy number. It should be verified.

5. Lines 124-125 – the conclusion that “Enok is important for proper transposon silencing in the germline” is not quite correct. Specifically, 7 TE families were significantly activated in enok mutants. Some kind of specificity is observed for Enok function and this fact should be discussed.

6. Lines 128-137. This paragraph is confusing. Can you explain why transcriptions of sense and antisense genomic strands in a subset of dual-strand clusters are regulated independently.

7. Lines 138-140 – definition of piRNA source loci and the difference between piRNA clusters and piRNA SL should be provided here. The differences in the number of piRNA clusters (140) and piRNA SL (several thousands) are especially confusing and have to be clearly explained.

8. Line 158 – the authors conclude that 20% of the Rhi-dependent loci are Enok-dependent (ED). Annotation of ED piRNA loci has to be performed in order to find similarities between these loci and to explain a possible mechanism of Enok targets selectivity (see also #15 comment).

9. Lines 177-178. Western blotting analysis is needed here to confirm reduction in the Rhi protein levels in the enok mutant ovaries.

10. Line 184. Specificity of a-Enok antibodies has to be verified by WB using ovary extracts from wild type and knockdown strains.

11. a-H3K23ac ChIP data (Fig S5b) are essential and would be better moved in the main figure.

12. Lines 251-253. A plausible explanation should be proposed to explain such various effects of enok mutations and knockdowns on expression of different targets, namely, 42AB and Rhi. In enok_KD, H3K23ac staining (Fig S7D) as well as H3K23ac enrichment at Rhi promoter (Fig. S5B) are significantly decreased. In lines 201-202 the authors state that “Enok may promote the expression of rhi and mael by acetylating H3K23”. Therefore, the situation is controversial, because enok_KD affects H3K23ac levels at Rhi promoter but does not affect Rhi levels. Can you clarify this? At least, H3K23ac enrichment at Rhi promoter has to be tested upon “mild” enok_KD.

13. The authors statement that “knocking down enok in the germline can decrease levels of RNAs from 42AB without affecting the global Rhi levels (Fig S7B-S7E)” is needed to be confirmed by more convincing experiments. Rhi-GFP protein levels have to be estimated more precisely using titration of the ovary extract and quantification. Loading control panel (Fig S7C) is overexposed that does not allow quantification. RT-qPCR of the Rhi RNA levels is highly recommended to back up the Rhi protein analysis in the ovaries of “mild” enok_KD.

14. Along the text, the expression “We have previously demonstrated…” should be replaced by ”it was previously demonstrated” because only one of the co-authors was involved in the mentioned studies before moving to another lab.

15. The authors discuss the regulation of piRNA-precursors transcription by Enok-mediated Rhi recruitment to the specific group of the germline piRNA clusters, however, no attempts were done to annotate the clusters related to this group. Recently, the specificity of the regulation of particular subsets of piRNA clusters, such as telomeric regions and euchromatic transposons, was demonstrated (Radion et al 2018, Kordyukova et al 2020). These data would be helpful for annotating Enok targets.

Reviewer #2: The last author of the present manuscript is also first author of a 2014 publication, which shows that the histone acetyltransferase Enok promotes expression of Spire and Maelstrom (Mael). Mael is involved in piRNA-mediated silencing of transposable elements (TEs) in Drosophila germline cells (Sienski et al. 2012). In the present manuscript, the authors were interested in a possible role of Enok in TE silencing. They examined TE transcription, production of small RNAs by piRNA clusters and by genome-wide 1-kb bins, they studied gene transcription levels, especially for genes involved in the piRNA pathway, as well as Enok, Rhino, and RNA polII occupancy on DNA, H3K23 acetylation, and Rhino, Mael and Piwi protein expression. These studies were done in Enok mutant germline clones, Enok-RNAi knock-down (KD) and control conditions.

Three transcriptome-wide RNAi screens identified genes, which are involved in the piRNA pathway in Drosophila, one screen for the germline and two for ovarian somatic cells (ref. 23, 25, 26 in the present manuscript). None of these screens did identify Enok as a possible candidate to be involved in the piRNA pathway and TE silencing. This discrepancy with the data presented here is discussed in the present manuscript (Lines 351-362). Still this discussion is based on unpublished data (Lines 356-357).

The manuscript is well written and the authors present here large amounts of data, obtained by RNA-seq, small-RNA-seq, ChIP-seq, ChIP-qPCR, RT-qPCR, Immunostaining and Western-blot, but unfortunately, the data as presented cannot be considered as conclusive. All statistical tests to check differential expression or IP seem to be simple T-Tests, for only 3 replicates of each condition, only 3 values being used for each T-test. Moreover, the correlation coefficients between many replicates are quite low (see comments). DESEQ or edgeR analyses will thus be required for all analyses of differential expression of transcripts (RNA-seq), small RNAs (small-RNA-seq), or of ChIP-Seq data. These more adequate statistical tools use the original non-normalized counts and are based on variance within the samples and between the samples. Simple T-tests are not sufficient to draw the right conclusions here. For gene expression, such analyses shall be done on all mRNAs, not only on genes of the piRNA pathway to be statistically consistent. The use of Wilcoxon Rank sum test for many of the presented data is not adequate in its implementation, allowing doubts on the drawn conclusions.

Concerning data obtained by qPCR, the primers present major defaults, several of them being non-specific, matching several times in the Drosophila genome presenting only 1 or 2 mismatches with off-target regions, other primers, supposed to be specific of one single locus, are so small, 11-13 nucleotides, that they match without mismatch up to 25 different loci in the Drosophila genome. The corresponding qPCR results, notably for piRNA cluster expression, cannot be conclusive (see also detailed comments).

In conclusion, in spite of the huge amount of data and possibly interesting results that may emerge if statistics are done correctly, the manuscript, at this point, is not suitable for publication. It needs major revision with profound changes concerning all statistics and primer design as suggested in my detailed comments before re-submission.

Major comments:

The authors never indicate which are the "Enok-dependent piRNA source loci". They correspond to specific 1-kb bins, but the authors do not indicate whether they are located in piRNA clusters, in the 42AB piRNA cluster, which is analysed in detail, or whether they are in totally different genomic regions, eventually even dispersed all over the genome. This important information remains obscure.

There is no link evidenced between piRNA cluster down regulation and the suggested up-regulation of 12 TE families. Actually, the authors do not show whether small RNAs complementary to these 12 TE families are down-regulated.

Primer design (Lines 515-561):

Several primers that were used to detect piRNA cluster or gene transcription are not specific (for example cl1-A-plus, cl1-32-plus, enok F). Indeed they map not only the locus aimed to be amplified but also elsewhere in the genome with only 1 or 2 internal or 5'-end mismatches (Flybase Blast, E=1000, no low complexity filter). Their 3'-end is not specific to the locus aimed to be amplified (see specific primer design for repeated sequences and its requirements in de La Roche Saint Andre C. and Bregliano J.C., Genetics 1998). Other primers are much too short to be specific (cl2-A-plus - 11 nucleotides, 25 matching loci in Release 6 genome; cl2-A-minus - 13 nucleotides, 11 matching loci in Release 6 genome) or contain palindromic sequences at their 3'-end (cl1-32-minus), presenting a risk of auto-amplification.

For primers destined to amplify TEs, it would be suitable to state whether they target specifically putative functional TEs or whether they may also amplify ancestral TE copies, which may be present in piRNA clusters. At least, it would be necessary to know whether the primers for TEs that seem to be de-silenced in enok mutant or KD conditions detect sequences within piRNA clusters, notably the 42AB cluster.

In several Figures appear p-values that have been calculated by a Wilcoxon Rank sum test. These p-values are extremely low and do not seem to be in adequacy with the corresponding box-plots. The Methods section gives details on how this test was implemented (Lines 446-452). It appears that the Wilcoxon Rank sum test has been done on means from 3 replicates for each 1-kb bin ("average numbers of reads from 3 independent replicates"). The authors have 3 replicates for each condition and they want to test whether medians of control samples are significantly different from the medians of mutant or knock-down (KD) samples. To do this a Kruskal-Wallis test will be needed. The results should then appear in a matrix and show no difference between replicates of the same condition, and significant difference between each control sample and each mutant or KD sample.

Lines 109-111: "Given the high degree of correlation between each independent replicate in our next generation sequencing (NGS) data sets (Fig S1 and S2), ..."

"High degree of correlation" may be true for RNA-seq data (Fig. S2A), but not for the other data, where, the correlation coefficients between many replicates are quite low, 2 out of 3 coefficients being <0.90 for all small-RNA-seq replicates (Figure S2B), 1 or 2 of 3 <0.90 for Enok ChIP-seq replicates (Figure S2C), and 1 out of 4 coefficients being <0.80 for replicates from RNA Pol II ChIP-seq (Figure S2D).

Lines 112-118:

I assume that Figure 1A shows all mapping reads, not only genome-unique mappers. Still this is not specified. Multiple-mappers may originate from other loci than the ones shown in the figure. Thus why do the authors present these specific loci for Burdock and HeT-A ? It is not even sure that these insertions exist in the genomic context used here.

One might assume that the data shown in Figure 1B and 1C correspond to RNA-seq data, as well as the data in "Table S1_TPMs of transposons.xlsx" but this is not specified. Moreover, Figures 1B and 1C show different results than Table S1.

Example: mean TPM from Table S1 for Burdock: wt - 1.1; enok1 - 21.6; enok2 - 16.9

Figure 1B,C show the following values for Burdock: wt - around 8; enok1 - around 100; enok2 - around 100

The same discrepancies are observed for other transposable elements in Figure 1B and 1C. In fact, the data in Figures 1B and 1C seem to be related to Table S1, but the scale is clearly erroneous (100 does clearly not correspond to 1 TPM, 101 does not correspond to 10 TPM etc.).

Table S1 ("Table S1_TPMs of transposons.xlsx"):

What do values like "9,6119E-115" (FBgn0001100_G-element) and "1,4906E-188" (FBgn0020425_Helena) mean, in a table supposed to show TPM (transcripts per million) ? Moreover, it is not clear what "TPM" means. How were these values calculated?

Table S2 ("Table S2_abundance of piRNAs mapping to piRNA clusters.xlsx"):

What values are given in this table, RPKM, RPM, normalized to all genome-unique reads, of which size ? The title of the table and the title of the sheet it contains suggest that these are genome-unique piRNA counts, but this is not specified in the legend or in the table. And again, as no Piwi-IP has been done to sort out piRNAs, what is the criterion to name these reads "piRNAs" ?

Lines 138-139:

It seems that the "piRNA source loci" correspond to 1-kb bins. How have these 1-kb bins been selected ? Are they contained in the 142 piRNA clusters identified by Brennecke et al., 2007 ? Even if this has been described in ref. 13, it is not clear to the reader whether the cutoff value of piRNA counts per bin was the same as in ref. 13. Indeed, this cutoff value should depend on the sequencing depth of each sample. Is this the case ? What about p-values to test differential production of small RNAs for single bins. It seems they were not considered here. Again, DESEQ analysis would be suitable. If the used cutoff value of piRNA counts per bin is too low, or variance too high, 1.5-fold changes may not be significant.

Lines 157-159: "Thus, we concluded that Enok may be important for piRNA production from ~20% of the Rhi-dependent loci."

This conclusion concerns 1-kb bins and nothing indicates whether these "20% of the Rhi-dependent loci" correspond to real "loci", for example piRNA clusters or whether they are dispersed over the genome. This would be essential to examine.

Lines 189-190: "Therefore, our results suggest that Enok may contribute to proper piRNA production partly by binding to rhi, mael and shu and facilitating their transcription."

The presented data do not clearly support this conclusion. It seems over-stated. Again DESEQ analyses on RNA-seq, small-RNA-seq and ChIP-seq might help to check whether the differentials are significant. These analyses should be done on all genes, not only on pre-selected genes known to be involved in piRNA biogenesis.

Lines 205-206: "Since rhi is severely down-regulated in enok mutant ovaries, ..."

This seems over-stated. Quantitative analyses are missing at this stage. Corresponding Western-blots are presented later in Figure S7 and their results are not concluding "severe down-regulation".

Lines 263-264: "This result suggests that Enok has a role in promoting transcription of a subset of dual-strand clusters including 42AB."

If the authors have arguments that "a subset of dual-strand clusters" is affected, why do they present only one dual-strand cluster ?

Lines 266-267: "Consistent with our small RNA-seq results (Fig 2C), Pol II occupancies at ED-SL were significantly reduced upon enok knockdown, while those at EI-SL were unaffected (Fig 5B)."

In addition to solving the problem of implementation of the Wilcoxon Rank sum test (see above), the authors should show data indicating which 1-kb bins have differential Pol II occupancies and whether they are located in piRNA clusters or grouped in other genomic loci.

Lines 323-327:

The authors propose a model of transposon regulation by Enok in Figure 7D. Unfortunately, in spite of the large quantity of data analysed here, it remains unclear how Enok "regulates the recruitment of Rhi to enhance transcription at piRNA clusters" as concluded by the authors. Moreover, the authors focus on the 42AB piRNA cluster but they do not state whether the transposable elements which seem to be de-silenced according to the presented data (Figure 1C) have related copies being present in the 42AB cluster and whether these copies lead to production of homologous regulating piRNAs in wt conditions.

Lines 358-359: "Czech et al. indeed showed that knocking down enok weakly activated the blood and Burdock transposons (less than 2-fold)."

I didn't find this information about fold-change in the cited publication (ref. 23). The corresponding z-scores in this publication were -0,505 and -0,740 respectively, thus non-significant.

Discussion:

Regarding the statistical analyses, which are not adequate, the discussion seems overstated and has clearly to be revised, once the data will be re-analysed using the adequate statistical tools and methods.

Minor comments:

There are not only "truncated transposons" (page 8 line 46) in Drosophila piRNA clusters, they may also contain full-length recent TEs. Please cite Zanni et al., 2013 here for the composition of a typical piRNA cluster.

The Piwi protein is not involved in PTGS, only in TGS. Thus the statement "Guided by complementary piRNAs, the Piwi-piRNA complex can mediate degradation of transposon transcripts' (Lines 51-52) is erroneous. The Piwi-piRNA complex enters the nucleus and induces TGS. What is true is that other PIWI-clade proteins "can mediate degradation of transposon transcripts" but not Piwi.

Lines 57-59:

The authors state " Transcription of the uni-strand clusters is proposed to be similar to the canonical transcription of protein-coding genes, as they contain clear promoter structures with enriched H3K4me2 and peaks of RNA polymerase II (Pol II) (13)."

I didn't find this information in the cited reference 13. Please verify.

Lines 89-91:

"Therefore, Enok contributes to proper transposon silencing in the germline both by activating canonical transcription of rhi and by promoting noncanonical transcription of piRNA clusters."

This conclusion is clearly overstated since non-canonical transcription is not evidenced by the data presented here.

Figure 1A:

Which is the unit for the y-axis in Figure 1A, RPKM (reads per kb per million), "TPM" as in Table S1 ? Normalized to genome-mapping reads, all reads, else ? As annotated in Release 6, the gene CG30345 overlaps Burdock in its 3'-UTR. This is not shown in the figure.

Line 130: "Among the 138 piRNA clusters in our analysis ..."

Why did the authors analyzed only 138 among the 142 piRNA clusters identified by Brennecke et al., 2007 ? Actually, the answer can be found in the legend to Figure 2. Please move this information to the main text.

Lines 143-144: "In contrast, piRNAs mapping to RI-SL, SO-SL or het non-SL were not down-regulated in enok mutants (Fig 2B)."

Were they up-regulated as the boxplots may suggest ?

Lines 170-172, Figure S4A:

Piwi seems to be up-regulated in enok1 mutant clones as shown in Figure S4A. This is not discussed.

Lines 172-173: "Immunostaining in the enok mutant germline clone ovaries also revealed reductions in the protein levels of Rhi and Mael (Fig S4C, S4D) similar to those in their mRNA levels (Fig S4A)."

Immunostaining showing one egg chamber is not sufficient to state this. Western blots are needed as a quantitative study. There is an analysis by Western blot for GFP-Rhino presented in Figure S7 but no Western-blot for Mael, in spite of the fact that anti-Mael antibodies were available.

Line 210:

In the results section, the authors indicate that the MTD-Gal4 driver is used for germline knock-down. The genotypes given in the figure legends do not mention MTD-Gal4. Please define MTD-Gal4 in the text to make the link with the figure legends.

Figure 3B:

The authors should clearly indicate the sense of transcription and the promoter regions or TSS of the different genes. Moreover, there are discrepancies between the presentation of exons, introns and CDS compared to the Release 6 annotation. The legend does not indicate anything about introns, exons and CDSs. The reader can only guess what boxes and lines correspond to. The authors should verify the gene representations in this figure.

The publication "Histone Acetyltransferase Enok Regulates Oocyte Polarization by Promoting Expression of the Actin Nucleation Factor Spire." (Fu Huang et al. Genes Dev, 28 (24), 2750-63, 2014) stated that "The expression of spir and mael are down-regulated in enok mutant ovaries". There is contradiction with the actual submitted manuscript, which concludes: "Figure S7. The protein levels of Rhi and Mael were largely unaffected in the enok knockdown ovaries."

There is essentially no discussion of this contradiction.

Line 591:

Reference 3 is not correctly cited. It should be Fejes Tóth K et al.

Comments on Methods:

Normalization for RNA-seq data is not specified. Normalization to million genome-mapping reads, or all reads, whether they map to the genome or not ?

The Drosophila Genome release used here was UCSC dm3 (Line 425, 442). Why not the latest release, which is release 6 ?

Which TE sequences were used, from which source ?

Lines 423-426: "The expression levels and the TPM values of transposon families were calculated with piPipes 1.5.0 ... (0 mismatch, multiple mappers assigned using EM algorithm) (27)".

What does that mean, random assignment of multi-mappers ?

Lines 412-413: "For examination of the mRNA levels of transposons or genes, qPCR was performed in triplicate for each sample, ..."

Are these biological (RNA extracts from different flies) or technical triplicates (from the same RNA preparation)? In the figure legends, the authors indicate that biological replicates, in general three, were used to determine p-values. Still, it is not clear whether there also were technical replicates for each RNA extract, as in common usage for qPCR, and how this was handled. Were the p-values calculated from means of technical triplicates, followed by T-tests on these means obtained for each biological replicate ?

Lines 426-427: "Transposon families activated in enok mutant ovaries were identified using the cutoff of P-value < 0.05, fold change > 2 and TPM > 1 in the enok1 data sets."

Why only enok1, not enok2 ?

How was determined this P-value, T-Test (Student) ?

Why didn't the authors use DESEQ analysis to identify up- or down-regulated transposon families ?

Line 431: "Total RNA was isolated from egg chambers of stages 5-14."

How did they isolate only stage 5-14 egg chambers ?

Lines 443-445: " The piRNA clusters with decreased or increased levels of piRNAs mapping to them in enok mutant ovaries were identified using the cutoff of P-value < 0.05."

Which test was used to determine these p-values?

Line 451: " P value of <2.2 x 10−16 is the maximum R can accurately calculate".

It is not clearly announced here that "R" was the tool used to calculate the p-values. Which "R" package and tool were used here ?

Lines 449 and 481: "bins with zero reads were excluded from the analysis."

It is not sufficient to exclude bins with 0 reads to determine significant fold-change. A higher threshold, statistically defined, should be applied.

Line 485 " Correlation Analysis of the NGS Data"

Which data were used as input files for this correlation analysis: multiBigwigSummary or multiBamSummary, 10-kb bins genome-wide ? Please specify.

Lines 501-502, Lines 511-513:

Antibodies should be denoted "anti-Piwi", "anti-Rhi" etc. for all of them, not only for "anti-H3K23ac".

Reviewer #3: piRNAs have a conserved role in transposon silencing, and germline piRNAs in Drosophila are derived from heterochromatic clusters marked by the HP1 homolog Rhino. The authors present evidence that the acetyl-transferase gene enok promotes rhino gene expression and independently enhances Rhino protein binding to a subset of piRNA clusters, increasing non-canonical transcription. The findings are potentially significant, but the phenotype is complicated and the data presentation and experimental strategy are somewhat convoluted, making interpretation difficult. As a result, it is unclear if enok or H3K23 have direct or indirect roles on rhi or piRNA cluster expression. Some straightforward experiments, using tools that appear to be in hand, could directly address these points and significantly improve the manuscript.

Specific recommendations:

1. Small RNAs.

The data for transposon over-expressed in germline clones of two alleles of enok is straightforward (Figure 1), but the small RNAseq data presentation in a series of Venn diagrams obscures key details. The cutoff points (fold change and statistical significance) for "up" and "down" regulated clusters are not clearly defined in the figure legend or text. Clusters are also very complex chromosomal domains, and both transcription and piRNA production varies widely with position. A single number does not capture this complexity, or changes in the pattern of long or short RNA expression. The authors should show genome browser screen shots of piRNAs uniquely mapping to both strands of representative clusters, and transposons. Scatter plots (best) or heat maps should then be used to present the data for all clusters and all transposon families. Binned data are a good option for clusters, and bins mapping to 42AB and cluster 2 could be highlighted in a different colors. A length histogram of total piRNAs, with ping-pong z-scores for clusters and transposons would also be useful. This would allow rapid assessment of global piRNA production, visual identification of changers in expression pattern across complex clusters, and would differentiate between clusters or transposon showing modest (2 fold, for example) or profound (100 fold) changes in expression.

The authors also present cluster-mapping piRNAs separated by strand, but how this was done is not defined. Clusters contain randomly oriented transposon fragments and are transcribed on both strands (42AB is typical), and genomic strand is not very meaningful for long or small RNAseq. Were cluster piRNAs broken down by strand relative to the inserted elements? If this was what was done, it needs to be stated. If not, strand separation is not very informative and piRNAs from both strands should be pooled.

2. rhino expression/H3K27ac

RNAseq data indicates the enok mutants reduce expression of rhino, and to a lesser extent mael and BoYb. ChIPseq shows that Enok associates with the promoters for these genes, suggesting that regulation may be direct. However, the Enok ChIPseq profiles in Figure S5 suggest that this protein binds to a very large number of genes. Does binding generally correlate with changes in expression on knock down? The direct output for Enok activity is presumably H3K27 acetylation, and assaying this modification is critical to the manuscript. The authors really need to perform H3K27ac ChIPseq, and show profiles across clusters, consensus transposons, and key genes. qPCR can be used to confirm the ChIPseq data, but these assays almost certainly miss key feature. These studies should also be done in the germline clones of chromosomal mutations, not with RNAi (see below).

3. Genetics.

The reliance on RNAi knock down for many of the critical studies is puzzling, and not explined in the text. The germline clones of characterized chromosomal mutations appear to produce robust phenotypes, and are not as inherently variable as knock down. These should be used for all sequencing, with RNAi as independent verification of the data.

4. Cluster transcription.

In Figure 4, the authors assay cluster transcription by qPCR, on RNAi knock down. However, Figure 1 presents RNAseq on germline clones of mutants. This is a much better way to define the impact of Enok on steady state cluster RNA expression, so date from the sequencing should be shown (browser shots, and scatter plots summarizing data for all clusters). Knock down and qPCR can be used to confirm the sequencing data. Similarly, Pol2 ChIPseq should be done on the germline clones. To confirm that changes in RNA and Pol2 are not due to reduced Rhino expression, the RNAseq and ChIPseq should be done in mutants over-expressing Rhino, as reported in the very nice rescue experiment in Figure S9E and F.

5. Interpretation.

The data presented in the paper indicate that enok mutants or RNAi reduce Rhino binding and expression of a subset of piRNA clusters. Transgenic over-expression nicely demonstrates that reduced Rhino protein is not responsible for reduced binding, suggesting that Enok directly promotes Rhino binding (Figure 7D). However, the authors show that Enok does not bind to piRNA clusters and may not change H3K27ac at these domains. Rhino binding is also reduced or eliminated by mutations in del, cuff, uap56, piwi, and THO complex components, other genes are also likely to impact this process, and enok mutations appear to disrupt expression of many genes. Together, these findings suggest that lack that Enok has an indirect role in promoting Rhino binding, likely through changes in expression of other genes.

Summary. The enok gene clearly has important functions in the germline, and histone acetylation has been largely ignored in studies on piRNA biogenesis and transposon silencing. The data presented here suggest that Enok and histone acetylation may be important for transcription of a subset of piRNA clusters. To convincingly address this possibility, and to rigorously characterize enok gene function, the authors should complete missing small RNA, long RNA, and ChIPseq experments, using control and chromosomal mutant strains, with and without Rhino over-expression. The data should be presented and analyzed in a way that clearly summarizes the findings across the genome and transcriptome.

**Have all data underlying the figures and results presented in the manuscript been provided?**

Reviewer #1: Yes

Reviewer #2: No: The data underlying figures concerning ChIP-seq, all raw read counts for RNA-seq, small-RNA-seq are missing. Read counts for the majority of genes, for all non-coding RNAs, read counts for 1-kb bins, classification of 1-kb bins in the different groups mentioned in the manuscript are missing. The only data underlying the figures that have been provided are normalized read counts for transposons (RNA-seq, Table S1), abundance of small RNAs mapping to piRNA clusters (Table S2) and normalized read counts of genes involved in the piRNA pathway (RNA-seq, Table S3).

Reviewer #3: Yes

PLOS authors have the option to publish the peer review history of their article (what does this mean?). If published, this will include your full peer review and any attached files.

Reviewer #1: No

Reviewer #2: No

Reviewer #3: No

---

## [Decision Letter · Decision Letter 1]

17 Aug 2020

Dear Dr Huang,

Thank you very much for submitting your Research Article entitled 'Acetyltransferase Enok regulates transposon silencing by promoting piRNA cluster transcription' to PLOS Genetics. Your manuscript was fully evaluated at the editorial level and by independent peer reviewers. The reviewers appreciate that the revisions you have made have improved the manuscript, but they still have some substantial concerns about the current manuscript, particularly with respect to claims extrapolated from the data and the interpretation of the results. Based on the reviews, we will not be able to accept this version of the manuscript, but we would be willing to review again a much-revised version in which you directly address these concerns. We cannot, of course, promise publication at that time.

If you decide to revise the manuscript for further consideration at PLOS Genetics, please aim to resubmit within the next 60 days, unless it will take extra time to address the concerns of the reviewers, in which case we would appreciate an expected resubmission date by email to plosgenetics@plos.org.

To resubmit, use 'Revise Submission' in the 'Submissions Needing Revision' folder.

We are sorry that we cannot be more positive about your manuscript at this stage. Please do not hesitate to contact us if you have any concerns or questions.

Yours sincerely,

Severine Chambeyron

Guest Editor

PLOS Genetics

Wendy Bickmore

Section Editor: Epigenetics

PLOS Genetics

Reviewer's Responses to Questions

**Comments to the Authors:**

Reviewer #1: The manuscript was considerably improved and most of the comments were thoroughly considered. The authors present convincing evidence that Enok promotes transcription of a subset of dual-strand clusters by facilitating Rhi recruitment. My major concern is related to the interpretation of the expression and sRNAseq data for TEs. Actually, enok depletion does not robustly affect TE piRNA levels while expression of several TEs is strongly upregulated. Moreover, chromatin changes and PolII occupancy at TEs upon Enok depletion were not considered in the paper. Data related to TEs do not fit the idea that Enok is essential for the piRNA cluster transcription and primary piRNA processing and THEREFORE for TE silencing (the Title and the model in Figure 7D). This straightforward explanation should be turned down because the story of TE silencing is more complicated than direct cluster-derived piRNA-mediated cleavage of TE mRNAs. An alternative explanation should be discussed. One may suggest that Enok directly or through Rhi assistance could affect TE RNA fate despite the presence of piRNAs. Anyway, the role of Enok in TE silencing is still unclear.

Specific comments:

1. The Title should be changed to not be so causative. “Acetyltransferase Enok regulates transposon silencing AND piRNA cluster transcription” is more relevant to the reported data.

2. The phrase (lines 148-149) “This result confirms that our small RNA-seq data are in line with the results from RNA-seq analysis (Fig 1C)” is over-statement and should be reformulate. It is the case only for Burdock and HeT-A but not for other upregulated TEs.

3. In this regard, scatter plot showing the levels of all TE piRNAs in enok mutants is required. Actually, piRNA reduction could explain upregulation of a few TEs. Likely, ping-pong mechanism with participation of maternal TE-specific piRNAs masks the decreased level of primary piRNAs in enok mutants. Enhanced RNA levels of TEs in enok mutants is likely determined by another mechanism.

4. In the present shape, Table S1, “TEtranscripts” list, is difficult to perceive the data. It would be more clear to the readers if TEs are arranged by fold change in RNA levels. The cutoff value (e.g. 1.0-log2FC) would be suitable to mark by color the affected TEs.

5. In Fig S4, the examples of piRNA clusters that are up-regulated in enok mutants should be present.

6. Figure S7C is important to demonstrate the role of Enok in Rhi expression and would be moved to Figure 3. However, co-immunostaining of Rhi and Piwi (or some other non-affected protein) is needed to be sure that weak Rhi staining in enok mutants is not an experimental failure.

7. Enok ChIP, Figure 3B: Negative control (no enrichment, no changes) and an example of non-piRNA pathway germline-specific gene enriched by Enok are required.

8. Figure 7D should be modified to show that Enok is essential i) for the transcription of Rhi; ii) for the Rhi-mediated transcription and primary piRNA processing of the cluster-derived piRNA precursors and iii) for TE silencing by unknown mechanism. An arrangement of arrows and brackets in this scheme, upper right, is confusing.

Reviewer #2: see attachment please

**Have all data underlying the figures and results presented in the manuscript been provided?**

Reviewer #1: Yes

Reviewer #2: **No: **It seems that the raw data (GSE105101) were not accessible to reviewers, even with the password for reviewers. The message was "External data have been provided but are not accessible for review while status is private." The reviewers seem to have access to the description of the sequencing samples only.

PLOS authors have the option to publish the peer review history of their article (what does this mean?). If published, this will include your full peer review and any attached files.

Reviewer #1: No

Reviewer #2: No

---

## [Editor Report · Decision Letter 2]

7 Jan 2021

Dear Dr Fu Huang,

We are pleased to inform you that your manuscript entitled "Acetyltransferase Enok regulates transposon silencing and piRNA cluster transcription" has been editorially accepted for publication in PLOS Genetics. Congratulations!

Yours sincerely,

Séverine Chambeyron

Guest Editor

PLOS Genetics

Wendy Bickmore

Section Editor: Epigenetics

PLOS Genetics

**Data Deposition**

http://datadryad.org/submit?journalID=pgenetics&manu=PGENETICS-D-20-00022R2

**Press Queries**

---

## [Editor Report · Acceptance letter]

25 Jan 2021

PGENETICS-D-20-00022R2 

Acetyltransferase Enok regulates transposon silencing and piRNA cluster transcription 

Dear Dr Huang, 

We are pleased to inform you that your manuscript entitled "Acetyltransferase Enok regulates transposon silencing and piRNA cluster transcription" has been formally accepted for publication in PLOS Genetics! Your manuscript is now with our production department and you will be notified of the publication date in due course.

With kind regards,

Alice Ellingham

PLOS Genetics

On behalf of:
